# The negative adipogenesis regulator *Dlk1* is transcriptionally regulated by *Ifrd1* (TIS7) and translationally by its orthologue *Ifrd2* (SKMc15)

Ilja Vietor[1]*[†], Domagoj Cikes[1,2][†], Kati Piironen[1,3][†], Theodora Vasakou[1][†], David Heimdörfer[4], Ronald Gstir[1,5][‡], Matthias David Erlacher[4], Ivan Tancevski[6], Philipp Eller[6][§], Egon Demetz[6], Michael W Hess[7], Volker Kuhn[8][#], Gerald Degenhart[9], Jan Rozman[10,11], Martin Klingenspor[12,13,14], Martin Hrabe de Angelis[10,11,15], Taras Valovka[1], Lukas A Huber[1,5]

[1]Institute of Cell Biology, Biocenter, Innsbruck Medical University, Innsbruck, Austria; [2]IMBA, Institute of MolecularBiotechnology of the Austrian Academy of Sciences, Vienna, Austria; [3]Division of Pharmaceutical Chemistry and Technology, Faculty of Pharmacy, University of Helsinki, Helsinki, Finland; [4]Division of Genomics and RNomics, Biocenter, Innsbruck Medical University, Innsbruck, Austria; [5]ADSI – Austrian Drug Screening Institute GmbH, Innsbruck, Austria; [6]Department of Internal Medicine II, Innsbruck Medical University, Innsbruck, Austria; [7]Division of Histology and Embryology, Innsbruck Medical University, Innsbruck, Austria; [8]Department Trauma Surgery, Innsbruck Medical University, Innsbruck, Austria; [9]Department of Radiology, Medical University Innsbruck, Innsbruck, Austria; [10]German Mouse Clinic, Institute of Experimental Genetics, Helmholtz Zentrum München, German Research Center for Environmental Health, Neuherberg, Germany; [11]German Center for Diabetes Research (DZD), Neuherberg, Germany; [12]Chair of Molecular Nutritional Medicine, Technical University of Munich, School of Life Sciences, Weihenstephan, Germany; [13]EKFZ - Else Kröner Fresenius Center for Nutritional Medicine, Technical University of Munich, Freising, Germany; [14]ZIEL - Institute for Food & Health, Technical University of Munich, Freising, Germany; [15]Chair of Experimental Genetics, Technical University of Munich, School of Life Sciences, Freising, Germany

**\*For correspondence:**
ilja.vietor@i-med.ac.at

[†]These authors contributed equally to this work

**Present address:** [‡]Division of Hygiene and Medical Microbiology at the Medical University of Innsbruck, Innsbruck, Austria; [§]Medical University of Graz, Department of Internal Medicine, Graz, Austria

[#]Deceased

**Competing interest:** The authors declare that no competing interests exist.

**Abstract** Delta-like homolog 1 (*Dlk1*), an inhibitor of adipogenesis, controls the cell fate of adipocyte progenitors. Experimental data presented here identify two independent regulatory mechanisms, transcriptional and translational, by which *Ifrd1* (TIS7) and its orthologue *Ifrd2* (SKMc15) regulate *Dlk1* levels. Mice deficient in both *Ifrd1* and *Ifrd2* (dKO) had severely reduced adipose tissue and were resistant to high-fat diet-induced obesity. Wnt signaling, a negative regulator of adipocyte differentiation, was significantly upregulated in dKO mice. Elevated levels of the Wnt/β-catenin target protein Dlk1 inhibited the expression of adipogenesis regulators *Pparg* and *Cebpa*, and fatty acid transporter *Cd36*. Although both *Ifrd1* and *Ifrd2* contributed to this phenotype, they utilized two different mechanisms. *Ifrd1* acted by controlling Wnt signaling and thereby transcriptional regulation of *Dlk1*. On the other hand, distinctive experimental evidence showed that Ifrd2 acts as a general translational inhibitor significantly affecting Dlk1 protein levels. Novel mechanisms of *Dlk1* regulation in adipocyte differentiation involving *Ifrd1* and *Ifrd2* are based on experimental data presented here.

## Editor's evaluation

This study provides important new insights into the molecular regulation of adipocyte differentiation. Two molecules, TIS7 and SKMc15, are shown to regulate the activity of the key transcriptional regulator DLK-1 via discrete mechanisms – one involving transcription and the other translation. These findings add valuable information to the well known roles of Wnt/catenin and PPARg on adipocyte differentiation and will provide an advance for those interested in the role of adipocytes in whole body metabolism.

## Introduction

Adipogenesis is a complex process in which multipotent stem cells are converted into preadipocytes before terminal differentiation into adipocytes (*Sarantopoulos et al., 2018*). These mechanisms involve protein factor regulators, epigenetic factors, and miRNAs. TPA-induced sequence 7 (TIS7) protein has been shown to be involved in the mainly transcriptional regulation of differentiation processes in various cell types, for example, neurons (*Iacopetti et al., 1996*), enterocytes (*Wang et al., 2005*), myocytes (*Vadivelu et al., 2004*), and also adipocytes (*Nakamura et al., 2013*).

Multiple lines of evidence link the regulation of Wnt/β-catenin signaling to the physiological function of *Tis7*, known in human as interferon-related developmental regulator 1 (*Ifrd1*) (*Iezaki et al., 2016*; *Vietor et al., 2005*). Experiments with *Ifrd1* knockout mice generated in our laboratory show a negative effect of *Ifrd1* on Wnt signaling and a positive effect on adipocyte differentiation (*Vietor et al., 2005*; *Yu et al., 2010*, #3955). *Ifrd1* deficiency leads to a significant upregulation of Wnt/β-catenin transcriptional activity in both primary osteoblasts and in mouse embryonic fibroblasts (MEFs) derived from *Ifrd1* knockout (KO) mice. It was shown that *Ifrd1* is also involved in the control of adipocytes differentiation in mice and was upregulated in both visceral white adipose tissue (vWAT) and subcutaneous white adipose tissue (sWAT) of genetically obese ob/ob mice (*Nakamura et al., 2013*). *Ifrd1* transgenic mice have increased total body adiposity and decreased lean mass compared with wild type (WT) littermates (*Wang et al., 2005*). On high-fat diet (HFD), *Ifrd1* transgenic mice exhibit a more rapid and proportionately greater gain in body weight with persistently elevated total body adiposity. Enhanced triglyceride (TG) absorption in the gut of *Ifrd1* transgenic mice (*Wang et al., 2005*) indicated that *Ifrd1* expressed in the gut epithelium has direct effects on fat absorption in enterocytes. As a result of impaired intestinal lipid absorption, *Ifrd1* KO mice displayed lower body adiposity (*Yu et al., 2010*). Compared with WT littermates, *Ifrd1* KO mice do not gain weight when chronically fed an HFD and *Ifrd1* deletion results in delayed lipid absorption and altered intestinal and hepatic lipid trafficking, with reduced intestinal TG, cholesterol, and free fatty acid mucosal levels in the jejunum (*Garcia et al., 2014*). Ifrd1 protein functions as a transcriptional co-regulator (*Micheli et al., 2005*) due to its interaction with protein complexes containing either histone deacetylases (HDAC) (*Park et al., 2017*; *Vadivelu et al., 2004*; *Vietor et al., 2002*; *Wick et al., 2004*) or protein methyl transferases, in particular PRMT5 (*Lammirato et al., 2016*). The analysis of adipocyte differentiation in preadipocytic 3T3-L1 cells suggested an involvement of *Ifrd1* in the regulation of adipogenesis in the Wnt/β-catenin signaling context (*Nakamura et al., 2013*).

SKMc15, also known as interferon-related developmental regulator 2 (*Ifrd2*), a second member of the *Ifrd* gene family, is highly conserved in different species (*Latif et al., 1997*). Mouse *Ifrd1* and *Ifrd2* are homologous, with a remarkable identity at both the cDNA and amino acid levels (58 and 88%, respectively). However, there was so far no information about the physiological function and mechanisms of action of Ifrd2 protein and its possible involvement in differentiation of various tissues. Recently, cryo-electron microscopy (cryo-EM) analyses of inactive ribosomes identified Ifrd2 as a novel ribosome-binding protein inhibiting translation to regulate gene expression (*Brown et al., 2018*). The physiological function of Ifrd2 matching the mechanism based on the abovementioned cryo-EM data was so far not shown. Ifrd2 could be involved in adipogenesis since a significant reduction of whole protein synthesis was previously shown as a major regulatory event during early adipogenic differentiation (*Marcon et al., 2017*).

Adipogenesis occurs late in embryonic development and in postnatal periods. Adipogenic transcription factors CCAAT/enhancer binding protein α (*Cebpa*) and peroxisome proliferator-activated receptor γ (*Pparg*) play critical roles in adipogenesis and in the induction of adipocyte markers (*Farmer, 2006*). *Pparg* is the major downstream target of Delta-like protein 1 (Dlk1). It is inactivated

by the induction of the MEK/ERK pathway, leading to its phosphorylation and proteolytic degradation (*Wang and Sul, 2009*). Dlk1, also known as Pref-1 (preadipocyte factor 1), activates the MEK/ERK pathway to inhibit adipocyte differentiation (*Kim et al., 2007*). *Cebpa* is highly expressed in mature adipocytes and can bind DNA together with Pparg to a variety of respective target genes (*Lefterova et al., 2008*). Besides, Pparg binding to *Cebpa* gene induces its transcription, thereby creating a positive feedback loop (*Lowell, 1999*). Both proteins have synergistic effects on the differentiation of adipocytes that requires a balanced expression of both *Cebpa* and *Pparg*.

Wnt/β-catenin signaling is one of the extracellular signaling pathways specifically affecting adipo-genesis (*Li et al., 2008*; *Ross et al., 2000*; *van Tienen et al., 2009*) by maintaining preadipocytes in an undifferentiated state through inhibition of *Cebpa* and *Pparg* (*Tontonoz and Spiegelman, 2008*). Pparg and Wnt/β-catenin pathways are regarded as master mediators of adipogenesis (*Xu et al., 2016*). Wnt signaling is a progenitor fate determinator and negatively regulates preadipocyte prolifer-ation through Dlk1 (*Mortensen et al., 2012*). Mice overexpressing *Dlk1* are resistant to HFD-induced obesity, whereas *Dlk1* KO mice have accelerated adiposity (*Moon et al., 2002*). *Dlk1* transgenic mice show reduced expression of genes controlling lipid import (*Cd36*) and synthesis (*Srebp1c*, *Pparg*) (*Barclay et al., 2011*). *Dlk1* expression coincides with altered recruitment of PRMT5 and β-catenin to the *Dlk1* promoter (*Paul et al., 2015*). PRMT5 acts as a co-activator of adipogenic gene expression and differentiation (*LeBlanc et al., 2012*). SRY (sex determining region Y)-box 9 (Sox9), a transcrip-tion factor expressed in preadipocytes, is downregulated preceding adipocyte differentiation. Dlk1 prevents downregulation of Sox9 by activating ERK, resulting in inhibition of adipogenesis (*Sul, 2009*). The PRMT5- and histone-associated protein Coprs affects PRMT5 functions related to cell differentia-tion (*Paul et al., 2012*). Adipogenic conversion is delayed in MEFs derived from *Coprs* KO mice and WAT of *Coprs* KO mice is reduced when compared to control mice. *Dlk1* expression is upregulated in *Coprs* KO cells (*Paul et al., 2015*).

Experimental data presented here show involvement of *Ifrd1* and *Ifrd2* in the process of adipocyte differentiation. dKO mice had strongly decreased amounts of the body fat when fed with even regular, chow diet and were resistant against the HFD-induced obesity. Two independent molecular mecha-nisms through which *Ifrd1* and *Ifrd2* fulfill this function were found. The fact that these two genes use independent mechanisms of action supported by the observation that whole-body deficiency of both genes led to a stronger phenotype when compared to single knockouts of *Ifrd1* or *Ifrd2*. *Ifrd1* regu-lates the Wnt signaling pathway activity and restricts Dlk1 protein levels, thereby allowing adipocyte differentiation. In contrast, *Ifrd2* KO did not affect Wnt signaling, but as we show here, cells lacking *Ifrd2* have significantly upregulated translational activity. In addition, strongly enriched *Dlk1* mRNA concentrations were identified specifically in polyribosomes isolated from *Ifrd2* knockout MEFs when compared to the WT MEFs. This was true also for dKO, but not for the single *Ifrd1* knockout cells. The ablation of both *Ifrd1* and *Ifrd2* genes significantly affected the expression of genes essential for adipocyte differentiation and function. Since dKO mice render a substantially leaner phenotype on chow diet, even without any challenge by HFD induction, *Ifrd1* and *Ifrd2* represent novel players in the process of physiological adipocyte differentiation.

## Results

### Mice lacking *Ifrd1* and *Ifrd2* genes have lower body mass, less fat, and are resistant against HFD-induced obesity

In order to clarify whether both *Ifrd1* and *Ifrd2* are involved in the regulation of the adipocyte differ-entiation and whether they act through the same or different mechanisms, mice lacking both genes were generated by crossing *Ifrd1* with *Ifrd2* single KO mice. dKO pups were viable, and adult male and female mice were fertile. At birth, body weights of dKO and WT mice were similar. Nevertheless, already during weaning, both the male and female dKO mice failed to gain weight when compared to their WT littermates and this persisted in the following weeks when the mice were fed regular diet (RD; chow diet) containing 11% kcal of fat. At 10 wk of age, dKO mice displayed 30% and later up to 44.9% lower body weight compared with WT mice (*Figure 1A*). In all further presented experiments, we used only male mice.

Based on dual-energy X-ray absorptiometry (DEXA) measurement, 6-month-old WT mice had substantially higher amounts of fat than their dKO littermates (*Figure 1B*, left panel). The effect of

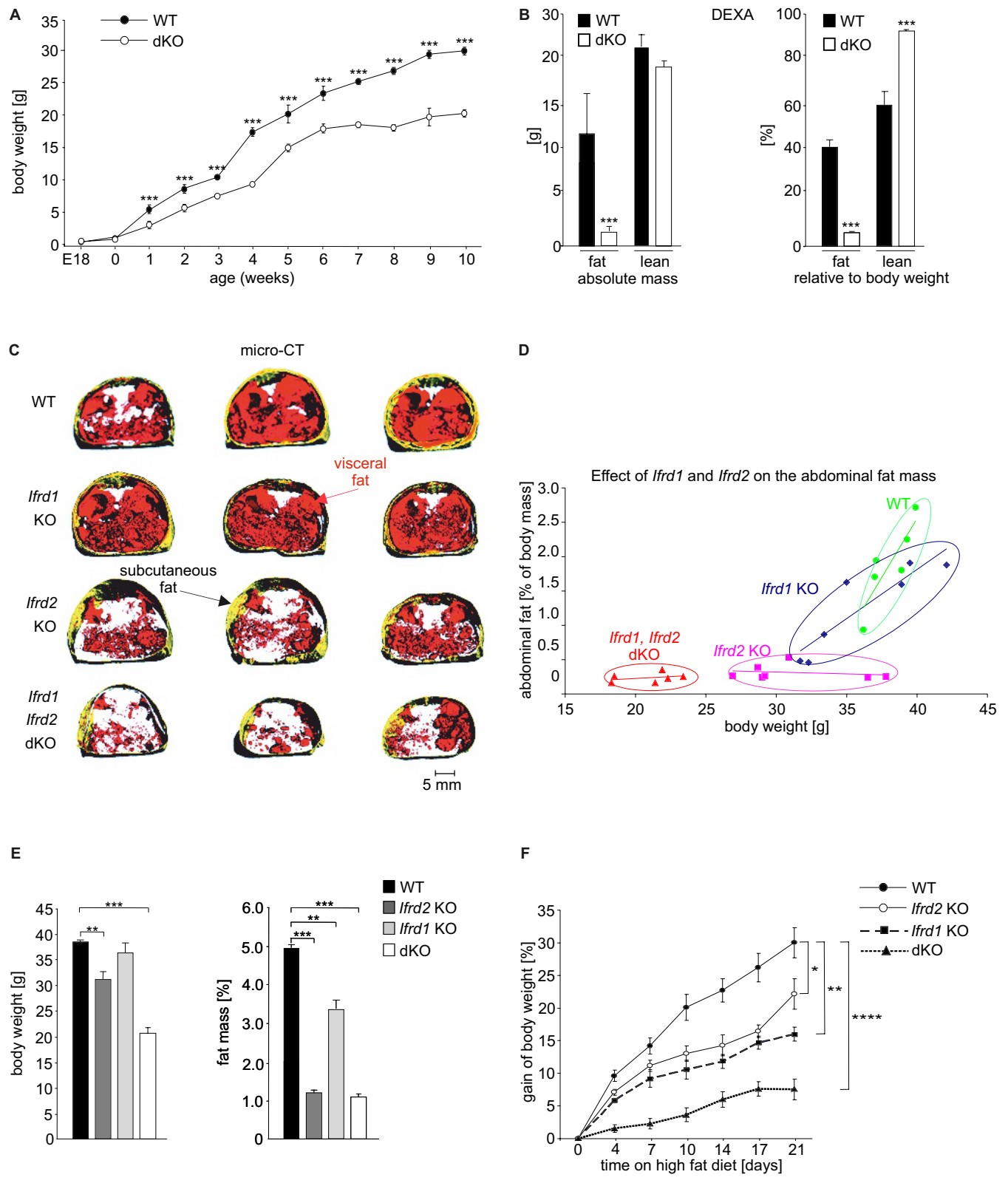

**Figure 1.** Double knockout (dKO) mice display postnatal growth retardation, less body fat, and are resistant against high-fat diet (HFD)-induced obesity. (**A**) Growth curves of wild type (WT) and dKO mice on chow diet (n fluctuate depending on the time point of measurement). (**B**) Dual-energy X-ray absorptiometry (DEXA) measurements of WT and dKO mice. Left panel depicts the absolute values of fat and lean mass per animal, and the right panel represents values normalized to the total body weight of animals (n = 15). (**C**) Micro-computed tomography measurements identified a lack of

*Figure 1 continued on next page*

*Figure 1 continued*

abdominal fat in single and dKO mice. Three-dimensional reconstitution of images of the abdominal fat mass distribution in WT and KO mice. Yellow color represents subcutaneous and red visceral fat mass. Black compartments are shadows resulting as part of the lightning model for the 3D volume rendering. (**D**) Mass contribution (%) of the abdominal fat in correlation to the total body weight (g). A linear regression is overlaid for each group individually. The regression results for the WT group are y = 0.342x - 11.13 with an $R^2$ of 0.74; for the *Ifrd1* KO group y = 0.143x - 3.91 with an $R^2$ of 0.82; for the *Ifrd2* KO y = 0.005x + 0.48 with an $R^2$ of 0.04 and; for the dKO y = 0.011x + 0.01 with an $R^2$ of 0.1. ANCOVA for the fat mass as a percentage of the body weight as covariant was performed. (**E**) Effect of single *Ifrd1* or *Ifrd2* knockout and *Ifrd1 Ifrd2* double knockout on total body weight in (g) and body fat amount normalized to the body weight in (%). (**F**) *Ifrd1* and *Ifrd2* knockout reduced the gain of body weight of mice fed with HFD. 7-week-old male WT and dKO mice were caged individually and maintained up to 21 d on HFD. Data shown are mean ± STD, n ≥ 9 per genotype. Data were analyzed applying one-way ANOVA with Holm–Šidák's multiple-comparisons test. *p<0.05, **p<0.01, ****p<0.0001.

The online version of this article includes the following source data and figure supplement(s) for figure 1:

**Source data 1.** DEXA and microCT measurements of IFRD1 and IFRD2 double knockout mice.

**Figure supplement 1.** Body length and respiratory exchange of double knockout (dKO) and wild type (WT) mice.

**Figure supplement 1—source data 1.** Further characterization of dKO mice.

**Figure supplement 2.** No difference between wild type (WT) and double knockout (dKO) mice in food consumption.

**Figure supplement 2—source data 1.** Supplementary information for the *Figure 1*.

*Ifrd1* and *Ifrd2* dKO was even more pronounced when the total fat and lean mass values were normalized to the body weight since the dKO mice were smaller than their WT counterparts were. The percentage of fat was in the WT mice 37.7 ± 4% vs. 6 ± 3% of the total body mass in dKO animals. Furthermore, the percentage of lean tissue mass in WT animals was lower than those in dKO animals (60 ± 4.6% vs. 92 ± 3.14%; *Figure 1B*, right panel). The dKO mice were slightly, but statistically significantly, smaller since there was a difference in body length, including the tail between WT and dKO mice (*Figure 1—figure supplement 1A*). Next, we analyzed the contribution of *Ifrd1* and *Ifrd2* to the whole-body fat content of mice. Three-dimensional reconstruction of images based on micro-computed tomography (micro-CT) of sex-/age-matched adult mice disclosed that both *Ifrd1* and *Ifrd2* single KO mice already had less abdominal fat than WT controls and that the dKO of *Ifrd1* and *Ifrd2* genes caused the strongest decrease in abdominal fat content and size (*Figure 1C*). Quantitative analyses of micro-CT measurements showed that *Ifrd1* deficiency caused a substantial lack of the abdominal fat tissues (p=0.002 when compared to WT mice, *Figure 1D*). Whereas *Ifrd1* KO mice had less fat mass but were not significantly lighter than their WT littermates (*Figure 1E*), *Ifrd2* KO were lighter (p=0.007) and leaner (p=0.0001) and dKO mice were both significantly lighter (p=0.0001) and had significantly less abdominal fat (p=0.006) than the WT mice (*Figure 1D and E*).

The indirect calorimetry trial with sex- and age-matched WT and dKO animals did not identify any significant difference in respiratory exchange ratios (RER = $VCO_2/VO_2$) of WT and dKO mice (*Figure 1—figure supplement 1B*). However, dKO mice showed significantly reduced body weight mainly because of lacking fat despite identical food intake, activity, and no major differences in several metabolic parameters. There were no statistically significant differences between WT and dKO mice even if the measured parameters were normalized to the smaller body mass (*Table 1*). To investigate potential links between *Ifrd1*, *Ifrd2*, and obesity, the response of dKO mice to HFD was studied. At 2 mo of age, male mice were housed individually and fed with HFD for 21 d. Food intake was measured every second day, and body weight was measured every fourth day. Feces were collected every second day to analyze the composition of excreted lipids, and blood samples were collected after the third week of HFD feeding to measure the concentrations of hepatic and lipoprotein lipases, respectively. Already within the first week of HFD feeding WT mice gained more weight than the dKO mice (*Figure 1F*), although there were no obvious differences in food consumption (*Figure 1—figure supplement 2A*) or in levels of lipolytic enzymes (*Figure 1—figure supplement 2B and C*). These differences in body weight gain continued to increase during the second and third weeks, at which time the body weight of WT mice increased additionally for 30% (30.0 ± 2.3%) when compared with the beginning of the HFD feeding period. In contrast, the weight of dKO animals increased only slightly (7.6 ± 1.6%) (*Figure 1F*). Both genes, *Ifrd1* and *Ifrd2*, contributed to this phenotype, and we could see a stronger effect following their deletion (*Figure 1F*).

**Table 1.** Double knockout (dKO) mice do not differ from wild type mice in any metabolic parameter besides the body weight.

A 21-hr indirect calorimetry trial monitoring gas exchange (oxygen consumption and carbon dioxide production), activity (distance and rearing), and food intake. The genotype effects were statistically analyzed using one-way ANOVA. Food intake and energy expenditure were analyzed using a linear model including body mass as a co-variate.

| | Wild type | dKO | ANOVA (genotype) * LM (body mass as co-variate) | |
| | n = 6 | n = 7 | Genotype | Body mass |
| Parameter | Mean ± STDEV | Mean ± STDEV | p Value | p Value |
|---|---|---|---|---|
| Body mass (g) | 31.1 ± 2.6 | 20.4 ± 1.9 | <0.0001 | n/a |
| Body temperature (°C) | 36.56 ± 0.6 | 36.71 ± 0.5 | 0.6515 | n/a |
| * Food intake (g) | 4.0 ± 0.4 | 3.4 ± 0.5 | 0.5869 | 0.1277 |
| * Mean VO$_2$ (ml/hr) | 107.28 ± 8.41 | 81.12 ± 5.69 | 0.6441 | 0.0241 |
| * Min VO$_2$ (ml/hr) | 78.83 ± 8.26 | 57.71 ± 8.86 | 0.9047 | 0.1219 |
| * Max VO$_2$ (ml/hr) | 144.33 ± 6.98 | 107.29 ± 7.59 | 0.1944 | 0.0082 |
| Mean RER | 0.89 ± 0.01 | 0.90 ± 0.02 | 0.3754 | n/a |
| Mean dist D (cm 20 min$^{-1}$) | 928 ± 173 | 874 ± 219 | 0.6399 | n/a |
| Mean Z (rearing 20 min$^{-1}$) | 116 ± 34 | 96 ± 25 | 0.2444 | n/a |

RER, respiratory exchange ratio.

## Adipocyte differentiation in dKO mice is inhibited due to upregulated DLK1 levels

A possible explanation of the lean phenotype of dKO mice was that *Ifrd1* and *Ifrd2* regulate adipocyte differentiation. Primary MEFs derived from totipotent cells of early mouse mammalian embryos are capable of differentiating into adipocytes and are versatile models to study adipogenesis as well as mechanisms related to obesity such as genes, transcription factors, and signaling pathways implicated in the adipogenesis process (*Ruiz-Ojeda et al., 2016*). To test whether *Ifrd1* and *Ifrd2* are required for adipogenesis MEFs derived from WT, *Ifrd2* and *Ifrd1* single and dKO mice were treated according to an established adipocyte differentiation protocol (*Wang et al., 2015*). Expression levels of both *Ifrd1* and *Ifrd2* mRNA increased during the adipocyte differentiation of WT MEFs. *Ifrd1* expression reached maximum levels representing 5.7-fold increase compared to proliferating WT MEFs on day 3 (*Figure 1—figure supplement 2D*) and *Ifrd2* reached on day 5 the maximum of 2.5-fold expression levels of proliferating MEFs (*Figure 1—figure supplement 2E*). These data suggested that both proteins play a regulatory role in adipogenesis; however, they differ in their mechanisms and timing. Eight days after initiation of adipocyte differentiation, a remarkable reduction of adipocyte differentiation potential in *Ifrd2*, *Ifrd1* KO MEFs, and dKO stromal vascular fraction (SVF) cells isolated from inguinal WAT was observed, as characterized by the formation of lipid droplets stained by oil red O (*Figure 2A*). Quantification of this staining revealed that fat vacuole formation in cells derived from *Ifrd2*, *Ifrd1,* and dKO mice represented 23, 48, and 12% of the WT cells, respectively (*Figure 2B*). Stable ectopic expression of *Ifrd2* significantly increased adipocyte differentiation in both single and double *Ifrd1* and *Ifrd2* knockout MEF cell lines (*Figure 1—figure supplement 1C and D* and *Figure 2—figure supplement 1*). Ectopic expression of *Ifrd1* significantly induced the adipocyte differentiation in *Ifrd1* single knockout MEFs (*Figure 1—figure supplement 1C*). These data indicated that both *Ifrd1* and *Ifrd2* were critical for adipocyte differentiation and that the defect in adipogenesis could be responsible for the resistance of dKO mice to HFD-induced obesity.

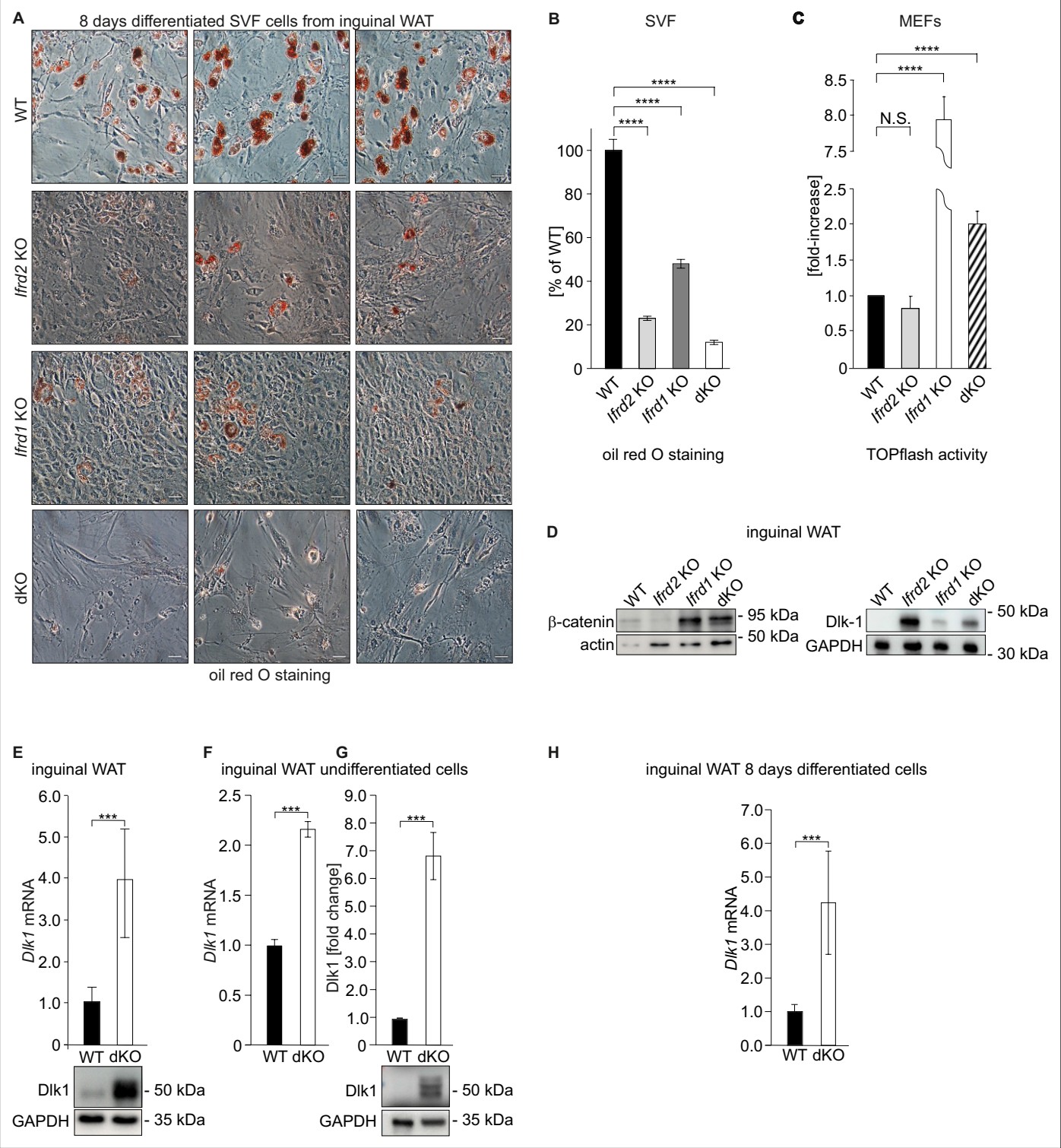

**Figure 2.** Adipocyte differentiation, *Dlk1* levels, and Wnt signaling pathway are significantly affected in double knockout (dKO) mice. (**A**) Representative oil red O staining of inguinal stromal vascular fraction (SVF) cells following 8 d of the adipocyte differentiation protocol. Scale bar is 20 μm. (**B**) Quantification of oil red O staining from three independent adipocyte differentiation experiments shown in panel (**A**). Ordinary one-way ANOVA ****p<0.0001. (**C**) Wnt signaling activity was measured in mouse embryonic fibroblasts (MEFs) transiently co-transfected with the Tcf reporter plasmid pTOPflash and β-galactosidase expression vectors. Transfection efficiency was normalized using the β-galactosidase values. The experiment was repeated three times. Data shown are mean ± STD. Ordinary one-way ANOVA Holm–Šídák's multiple-comparisons test ****p<0.0001. (**D**) Left top panel: western blot analysis of β-catenin amounts in inguinal WAT samples. Representative western blot from three biological repetitions is shown. Right top

*Figure 2 continued on next page*

*Figure 2 continued*

panel: Dlk1 is significantly upregulated both in *Ifrd1*, *Ifrd2* single KO and in dKO mice. Bottom panels: actin western blots were used for normalization of sample loading. (**E**) *Dlk1* mRNA expression measured by qPCR in inguinal white adipose tissue (WAT) samples (top). Values were normalized on *GAPDH*, n = 3. Error bars indicate standard deviations. A representative western blot image detecting *Dlk1* and *GAPDH* proteins (bottom). (**F**) *Dlk1* mRNA expression detected in undifferentiated cells isolated from inguinal WAT SVF cells. Normalized on *GAPDH*. Error bars indicate standard deviations, ***p<0.001. (**G**) Dlk1 protein western blot analysis in undifferentiated cells isolated from inguinal WAT SVF cells. Mean of three biological repeats; inset is one representative western blot. ***p<0.001. (**H**) *Dlk1* mRNA expression detected in 8 d adipocyte-differentiated cells isolated from inguinal WAT SVF cells. Normalized on *GAPDH*. Error bars indicate standard deviations, ***p<0.001.

The online version of this article includes the following source data and figure supplement(s) for figure 2:

**Source data 1.** b-catenin and Dlk1 protein levels in inguinal WAT.

**Source data 2.** Dlk1 mRNA levels in inguinal WAT.

**Source data 3.** Dlk1 protein levels in inguinal WAT cells.

**Figure supplement 1.** *Ifrd1* and *Ifrd2* positively regulate adipocyte differentiation of mouse embryonic fibroblast (MEF) cells and inhibit *Dlk1* expression.

Wnt/β-catenin signaling is an important regulatory pathway for adipocyte differentiation (***Prestwich and Macdougald, 2007***). *Ifrd1* deficiency causes upregulation of Wnt/β-catenin transcriptional activity. Therefore, Wnt signaling activity was assessed in MEFs by measuring transcriptional activity using TOPflash, TCF-binding luciferase reporter assays. As shown in ***Figure 2C***, Wnt signaling activity, when compared to the WT MEFs, was not regulated in *Ifrd2* KO, but highly significantly (p<0.0001) upregulated in *Ifrd1* KO and also in dKO MEFs. Supporting these findings, western blot analyses also identified increased β-catenin protein levels in inguinal WAT of *Ifrd1* KO and in dKO mice (p<0.0001), but not in *Ifrd2* KO mice (***Figure 2D***, left top panel), suggesting the involvement of *Ifrd1* but not *Ifrd2* in the Wnt signaling pathway regulation.

Next, the expression levels of *Dlk1*, a negative regulator of adipogenesis and at the same time known target of Wnt signaling (***Paul et al., 2015***), were analyzed. Dlk1 protein was significantly upregulated in inguinal WAT isolated from *Ifrd1* and *Ifrd2* KO as well as from dKO mice (***Figure 2D***, right top panel). A significant (p<0.001) upregulation of *Dlk1* mRNA and protein levels in dKO inguinal WAT samples revealed the qPCR and confirmed western blot analyses (***Figure 2E***). Upregulation of *Dlk1* in dKO mice both on RNA and protein results was confirmed in undifferentiated SVF cells isolated from inguinal fat (***Figure 2F and G***). *Dlk1* mRNA levels were even stronger upregulated following the 8-days differentiation protocol of SVF cells (***Figure 2H***). Moreover, whereas in WT MEFs the expression of *Dlk1* was strongly upregulated only during the first day of adipocyte differentiation and then over the next days declined to basal levels, in dKO MEFs *Dlk1* was upregulated (p<0.001) throughout the entire 8 d of differentiation (***Figure 3A***). This result complemented protein analyses of lysates from 8-day differentiated adipocytes (***Figure 3—figure supplement 1***, middle panel). A rescue experiment confirmed that *Dlk1* expression was *Ifrd1*- and *Ifrd2*-dependent. *Dlk1* mRNA levels were analyzed by RT-qPCR in dKO MEFs stably expressing *Ifrd1* and/or *Ifrd2*. Ectopic expression of *Ifrd1*, *Ifrd2* and mainly their combination significantly (p<0.001) downregulated *Dlk1* mRNA and protein levels (***Figure 3B***). Accordingly, these experiments documented that *Ifrd1* and/or *Ifrd1* were involved in the regulation of *Dlk1* expression, but the molecular mechanism remained unclear.

The adipocyte differentiation deficiency of dKO inguinal SVF cells suggested that *Cebpa* and *Pparg* might also be regulated through Ifrd1 and/or Ifrd2. It was previously shown that elevated levels of the cleaved ectodomain of *Dlk1* have been correlated with reduced expression of *Pparg* (***Lee et al., 2003***). Therefore, the differences in *Pparg* expression between WT and dKO inguinal SVF cells were analyzed. While *Pparg* and *Cebpa* mRNA levels were strongly induced in undifferentiated WT, these were barely detectable in dKO SVF cells (***Figure 3C and D***). Similarly, significantly decreased *Pparg* and *Cebpa* mRNA levels in *Ifrd1*, *Ifrd2*, and dKO inguinal SVF cells following 8-day adipocyte differentiation protocol were identified (***Figure 3E and F***).

Earlier chromatin immunoprecipitation (ChIP) analysis revealed that Ifrd1 binds directly to DNA and via interaction with PRMT5 regulates gene expression (***Lammirato et al., 2016***). Therefore, ChIP experiments were performed to study binding of Ifrd1 and Ifrd2 proteins, transcription factor β-catenin, and symmetrically dimethylated histone H4 at arginine residue 3 (H4R3me2s) (***Paul et al., 2015***) to regulatory elements of the *Dlk1* gene. dKO MEFs treated 8 d with the adipocyte differentiation cocktail were increased β-catenin binding to the β-catenin/TCF binding site 2 of the *Dlk1* regulatory

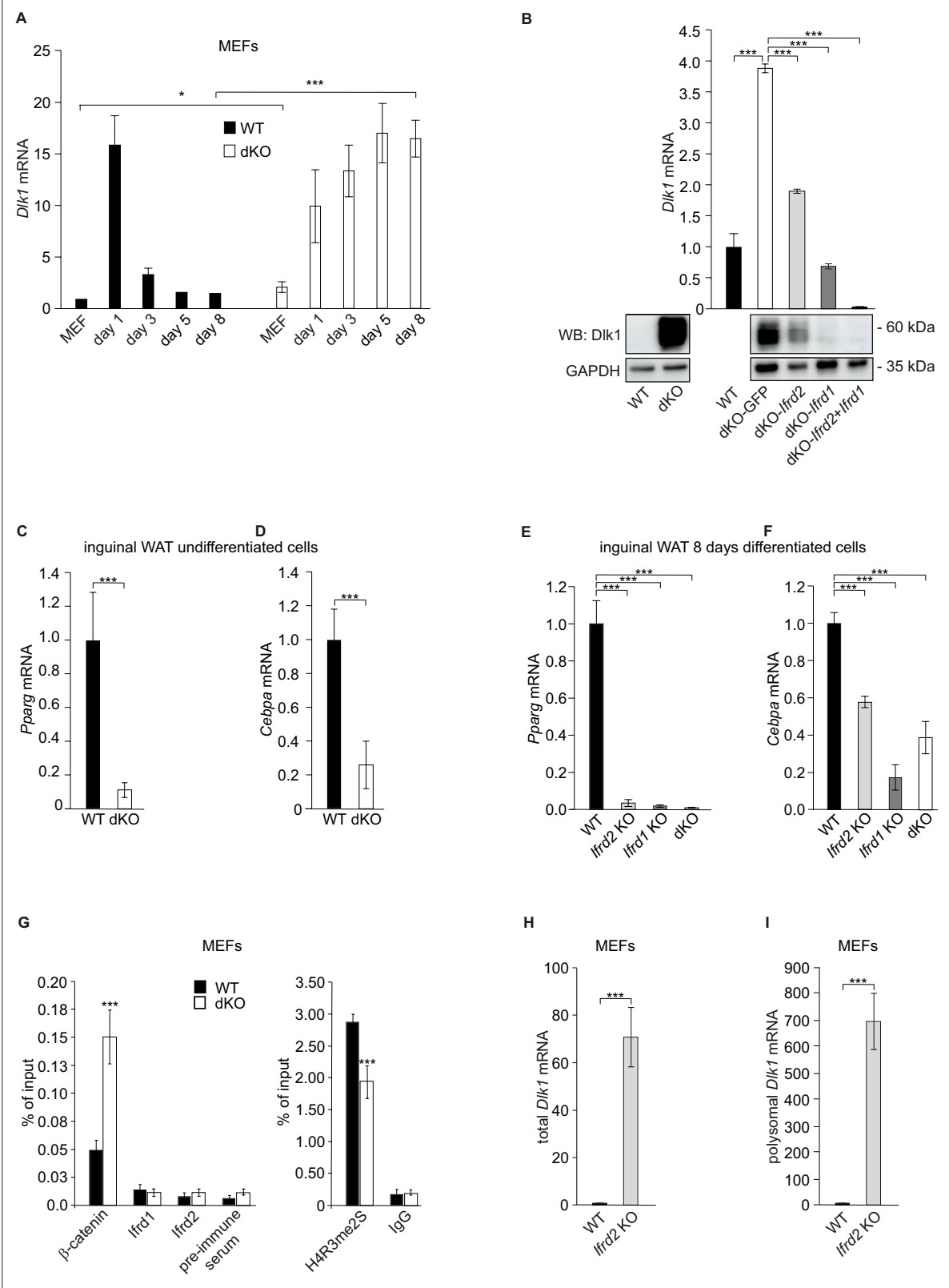

**Figure 3.** *Ifrd1* and *Ifrd2* regulate *Dlk1* and thereby affect adipocyte regulators expression. (**A**) *Dlk1* mRNA expression measured in mouse embryonic fibroblasts (MEFs) treated with the adipocyte differentiation cocktail for given times. Wild type (WT) MEF values were set as 1. *p<0.05, ***p<0.001. (**B**) *Dlk1* mRNA levels (top) and protein levels (bottom) in double knockout (dKO) MEFs were downregulated following ectopic co-expression of *Ifrd1* and *Ifrd2*. mRNA expression levels were analyzed by RT qPCR in stably transfected cells following 8-day adipocyte protocol differentiation. WT

*Figure 3 continued on next page*

*Figure 3 continued*

MEF values were set as 1, n = 3; error bars indicate standard deviations. ***p<0.001 (**C**) *Pparg* mRNA expression detected in undifferentiated cells isolated from inguinal white adipose tissue (WAT) stromal vascular fraction (SVF) cells. Normalized on *GAPDH*. Error bars indicate standard deviations, ***p<0.001. (**D**) *Cebpa* mRNA expression detected in undifferentiated cells isolated from inguinal WAT SVF cells. Normalized on *GAPDH*. Error bars indicate standard deviations, ***p<0.001. (**E**) *Pparg* mRNA expression detected in 8-day adipocyte-differentiated cells isolated from inguinal WAT SVF cells. Normalized on *GAPDH*. Error bars indicate standard deviations, ***p<0.001. (**F**) *Cebpa* mRNA expression detected in 8-day adipocyte-differentiated cells isolated from inguinal WAT SVF cells. Normalized on *GAPDH*. Error bars indicate standard deviations, ***p<0.001. (**G**) Recruitment of indicated proteins to regulatory regions of the *Dlk1* promoter in WT and KO MEFs was analyzed by chromatin immunoprecipitation (ChIP) at day 8 of the adipocyte differentiation. Values are expressed as the percentage of immunoprecipitated chromatin relative to input and are the mean of triplicates. ChIP analysis identified increased specific β-catenin binding to its *Dlk1* regulatory element in dKO samples. Pre-immune serum and IgG were used as background controls. n = 3, data shown are mean ± STD, Student's *t*-test ***p<0.001. (**H**) *Dlk1* mRNA expression detected in undifferentiated MEF cells. Normalized on *GAPDH*. Error bars indicate standard deviations, ***p<0.001. (**I**) Real-time qPCR detection of *Dlk1* RNA in polyribosomes. Normalized on *GAPDH*. Error bars indicate standard deviations, ***p<0.001.

The online version of this article includes the following source data and figure supplement(s) for figure 3:

**Source data 1.** Supplementary information for the *Figure 3* - WB images.

**Source data 2.** qPCR measurements for *Figure 3G*.

**Source data 3.** Supplementary information for the *Figure 3*.

**Figure supplement 1.** *Dlk1* inhibits adipocyte differentiation of double knockout (dKO) mouse embryonic fibroblasts (MEFs).

**Figure supplement 1—source data 1.** WB images for the *Figure 3—figure supplement 1*.

element found when compared to WT MEFs (*Figure 3G*). On the other hand, binding of H4R3me2s to the same *Dlk1* regulatory element was significantly reduced (p<0.001). No direct binding of Irfd1 or Ifrd2 proteins to two different *Dlk1* regulatory elements (*Dlk1* region A and β-cat/TCFbs2), neither in WT nor in dKO MEFs, could be identified, suggesting rather an epigenetic regulation than via their direct binding to *Dlk1* regulatory elements. These results suggested that proteins Ifrd1 and Ifrd2 are required to restrain the *Dlk1* levels through the Wnt/β-catenin signaling pathway and yet another so far unknown mechanism.

After finding significantly increased Dlk1 protein levels in inguinal WAT of SKMc15 single knockout mice (*Figure 2D*), *Dlk1* expression in MEFs generated from these mice was measured. These were significantly increased (>70-fold) when compared to WT MEFs (*Figure 3H*). In a search for a Ifrd2-specific regulatory mechanism of *Dlk1* levels, primarily the translational regulation was studied. It was previously shown that general reduction of protein synthesis and downregulation of the expression and translational efficiency of ribosomal proteins are events crucial for the regulation of adipocyte differentiation (*Marcon et al., 2017*). Initially, specifically polyribosome-bound *Dlk1* RNA in *Ifrd2* knockout MEFs was measured. This analysis detected significantly (p<8.26104 E-07) higher *Dlk1* mRNA levels in polyribosome RNA fraction of *Ifrd2* KO when compared to WT MEFs (*Figure 3I*). Interestingly, *Ifrd2* was recently identified as a novel specific factor capable of translationally inactivating ribosomes (*Brown et al., 2018*). Because *Ifrd2* may play a crucial role in the adipogenesis regulation in the following experiment, the effect of *Ifrd2* knockout on the general translational efficiency of WT, *Ifrd1*, *Ifrd2* single and double KO MEFs was tested. Cells were incubated 30 min in the absence of methionine and cysteine, followed by 1 hr in the presence of $^{35}$S-methionine. As shown in *Figure 4A*, there was a significant increase in the general translational activity of MEFs lacking *Ifrd2* alone or both *Ifrd1* and *Ifrd2*, but not in TIS7 single knockout cells. These data suggested that *Ifrd2* alone, but not *Ifrd1*, inhibits the general translational activity necessary for the induction of adipogenic differentiation, also via the translational regulation of *Dlk1*. This finding supported the result shown in *Figure 2D* where the Dlk1 protein levels were significantly induced in inguinal WAT samples isolated from *Ifrd2* single knockout mice.

## TIS7 and SKMc15 regulate adipocyte differentiation through Dlk1, MEK/ERK pathway, *Pparg*, and *Cebpa*

Dlk1 protein carries a protease cleavage site in its extracellular domain (*Lee et al., 1995*) and is secreted. The extracellular domain of Dlk1 is cleaved by ADAM17, TNF-α converting enzyme to generate the biologically active soluble Dlk (*Wang and Sul, 2009*). *DLK1* mRNA and protein levels are high in preadipocytes, but *Dlk1* expression is absent in mature adipocytes. Hence, adding soluble Dlk1 to the

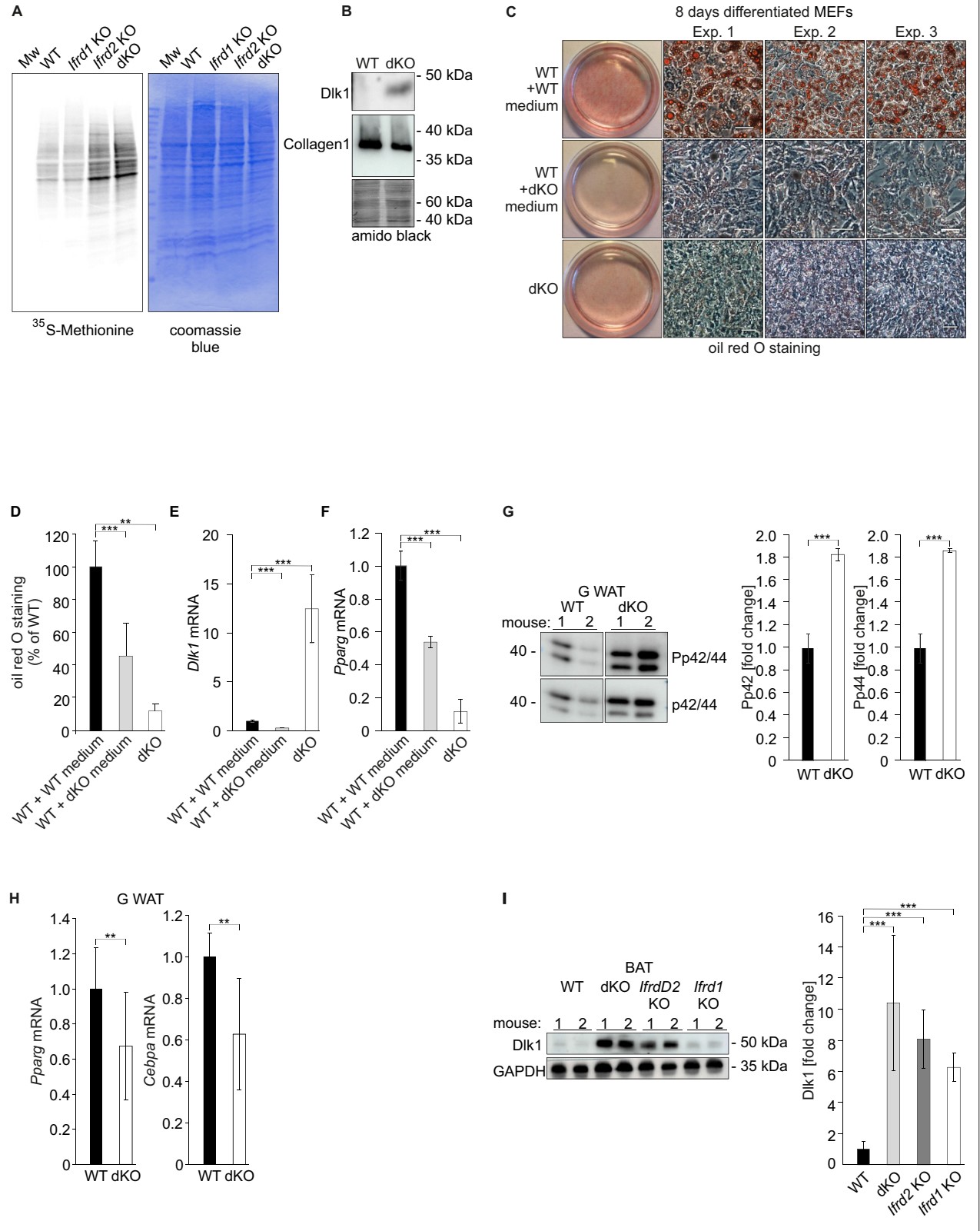

**Figure 4.** *Ifrd2* inhibits *Dlk1* translation. Double knockout (dKO)-secreted *Dlk1* inhibits adipocyte differentiation through MEK/ERK signaling. (**A**) Translational analysis – in vivo metabolic staining of mouse embryonic fibroblasts (MEFs) with <sup>35</sup>S-methionine. Equal numbers of cells were seeded and 24 hr later treated as explained in the 'Methods' section. Identical volumes of cell lysates were separated by SDS-PAGE, gel was dried and analyzed by a phosphorimager. Equal loading was documented by the Coomassie blue staining of the gel. (**B**) dKO MEFs secrete Dlk1 protein into the

*Figure 4 continued on next page*

*Figure 4 continued*

cell culture medium. Identical volumes of media from wild type (WT) and dKO MEF cells 8 d treated with the adipocyte differentiation cocktail were analyzed by western blot. Collagen I was present in both samples in similar amounts. Equal loading of samples was evaluated by amido black staining of the membrane. (**C**) dKO MEFs-conditioned medium inhibited adipocyte differentiation in WT MEFs. MEFs were treated 8 d with the adipocyte differentiation cocktail. WT MEFs treated with the dKO-conditioned medium (replaced three times every 2 d) showed reduced differentiation. Images representing three biological repeats were cropped, and space bars represent always 20 μm. (**D**) Quantification of oil red O staining from three independent adipocyte differentiation experiments. Data shown are mean ± STD, ordinary one-way ANOVA p=0.0016. Student's *t*-test, **p<0.01, ***p<0.001. (**E**) *Dlk1* mRNA levels in WT and dKO MEFs treated 8 d with the adipocyte differentiation cocktail or in WT MEFs treated with the dKO-conditioned medium, ***p<0.001. (**F**) *Pparg* mRNA expression levels in same cells as shown in panels (**D**) and (**E**), ***p<0.001. (**G**) Representative western blots of phospho-p44, phospho-p42, p44, and p42 in gonadal white adipose tissue (WAT) of WT and dKO mice. Normalization on p44 and p42; n = 3. Error bars indicate standard deviations. WT values were set as 1, ***p<0.001. (**H**) *Pparg* and *Cebpa* mRNA expression in gonadal WAT. Normalization on *GAPDH*; n = 3. Error bars indicate standard deviations. WT values were set as 1, **p<0.01. (**I**) Dlk1 protein is upregulated in brown adipose tissues (BAT) of *Ifrd1*, *Ifrd2* single knockout and in dKO mice. Western blot analysis was performed on five samples of each genotype. Normalization on *GAPDH*. Error bars indicate standard deviations, ***p<0.001.

The online version of this article includes the following source data and figure supplement(s) for figure 4:

**Source data 1.** Original images of the *Figure 4A*.

**Source data 2.** Original images of the *Figure 4B*.

**Source data 3.** Original images of the *Figure 4G*.

**Source data 4.** Original images of the *Figure 4I*.

**Figure supplement 1.** *Dlk1* inhibits adipocyte differentiation of double knockout (dKO) mouse embryonic fibroblasts (MEFs) and regulates *Hes1* expression.

**Figure supplement 1—source data 1.** Original images of the *Figure 4—figure supplements 1 and 2*.

**Figure supplement 2.** *Dlk1*, adipocyte differentiation of double knockout (dKO) mouse embryonic fibroblast (MEF) cells, and *Cebpa* expression.

medium inhibits adipogenesis (*Garcés et al., 1999*). To test whether the dKO MEFs secreted Dlk1, cell culture media from MEFs treated 8 d with the adipocyte differentiation cocktail were collected and analyzed by western blotting. As shown in *Figure 4B*, dKO cells secreted Dlk1 protein, but in an identical volume of the cell culture medium from WT MEFs no Dlk1 could be detected. In contrast, an unrelated secreted protein, namely collagen I, was found in media of both WT and dKO MEFs in similar, in the medium of WT MEFs even slightly higher, amounts. To prove that secreted *Dlk1* could inhibit adipocyte differentiation of dKO MEFs, WT MEFs were cultured with conditioned medium from dKO MEFs. Adipocyte differentiation of WT MEFs was strongly inhibited by the dKO MEFs-conditioned medium when compared to the control WT cells (*Figure 4C*, quantified in *Figure 4D*). Another indication that dKO inhibited adipocyte differentiation via Dlk1 protein upregulation delivered the experiment where *Dlk1* was specifically knocked down. Targeted were either all *Dlk1* mRNA splice variants (oligo sh*DLK1* 391) or only *Dlk1* mRNA splice variants containing coding sequences for the protease site for extracellular cleavage (oligo sh*DLK1* 393) as previously published in *Mortensen et al., 2012*. Both *Dlk1* knockdown constructs stably expressed in dKO MEFs significantly increased (p<0.001) adipocyte differentiation as documented in *Figure 4—figure supplement 1A*. It is known from the literature that *Hes1* levels, together with *Dlk1,* are continuously downregulated during the process of adipogenesis while *Pparg* are rising (*Huang et al., 2010*). The knockdown of *Dlk1* levels was paralleled by a significant (p<0.001) decrease in *Hes1* mRNA levels (*Figure 4—figure supplement 1B*). In contrast, treatment with a recombinant Dlk1 protein or stable *Dlk1* ectopic expression documented by qRT-PCR (*Figure 4—figure supplement 2C*) significantly (p<0.001) inhibited adipocyte differentiation of WT MEFs as shown in *Figure 4—figure supplement 2A and B*. This was accompanied by a significant (p<0.001) decrease in *Cebpa* mRNA levels (*Figure 4—figure supplement 2D*). Furthermore, *Dlk1* mRNA quantification (*Figure 4E*) documented that dKO MEF cell lysates contained significant amounts of *Dlk1* mRNA when compared to WT control or WT cells treated with dKO MEFs-conditioned medium. In parallel, WT cells treated with dKO MEFs-conditioned medium expressed significantly lower amounts of *Pparg* mRNA when compared to WT cells incubated with control medium (*Figure 4F*). In addition, ectopic expression of *Ifrd2* and co-expression with *Ifrd1* in dKO MEFs rescued almost up to WT levels the adipocyte differentiation potential of these cells (*Figure 2—figure supplement 1*). Ectopic expression of both *Ifrd1* and *Ifrd2* significantly (p<0.001) downregulated *Dlk1* mRNA expression in dKO MEFs (*Figure 2—figure supplement 1*). Moreover, conditioned medium from dKO MEFs expressing *Dlk1* shRNA knockdown constructs significantly

(p<0.001) lost the ability to inhibit adipocyte differentiation of WT MEFs as the medium from dKO MEFs did (*Figure 4—figure supplement 2E*). These results revealed that cells derived from dKO mice express increased *Dlk1* levels and secreted; soluble Dlk1 may inhibit adipocyte differentiation in vivo.

Previous studies showed that soluble Dlk1 protein activates MEK/ERK signaling, which is required for inhibition of adipogenesis (*Kim et al., 2007*). As *DLK1* was strongly upregulated in dKO SVF cells during adipocyte differentiation, the possible activation of the MEK/ERK pathway in gonadal WAT samples of WT and dKO mice was analyzed subsequently (*Figure 4G*). The phosphorylation of p44 and p42 was upregulated 1.8-fold (p<0.001) in the dKO when compared to the WT G WAT samples (*Figure 4G*). Next, the question of expression levels of *Pparg* and *Cebpa* in WAT depots was addressed. Both adipocyte differentiation regulators *Pparg* and *Cebpa*'s mRNA expression levels were in gonadal WAT samples isolated from dKO mice significantly (p<0.01) downregulated when compared to the values of WT control animals (*Figure 4H*). Furthermore, the possible effect

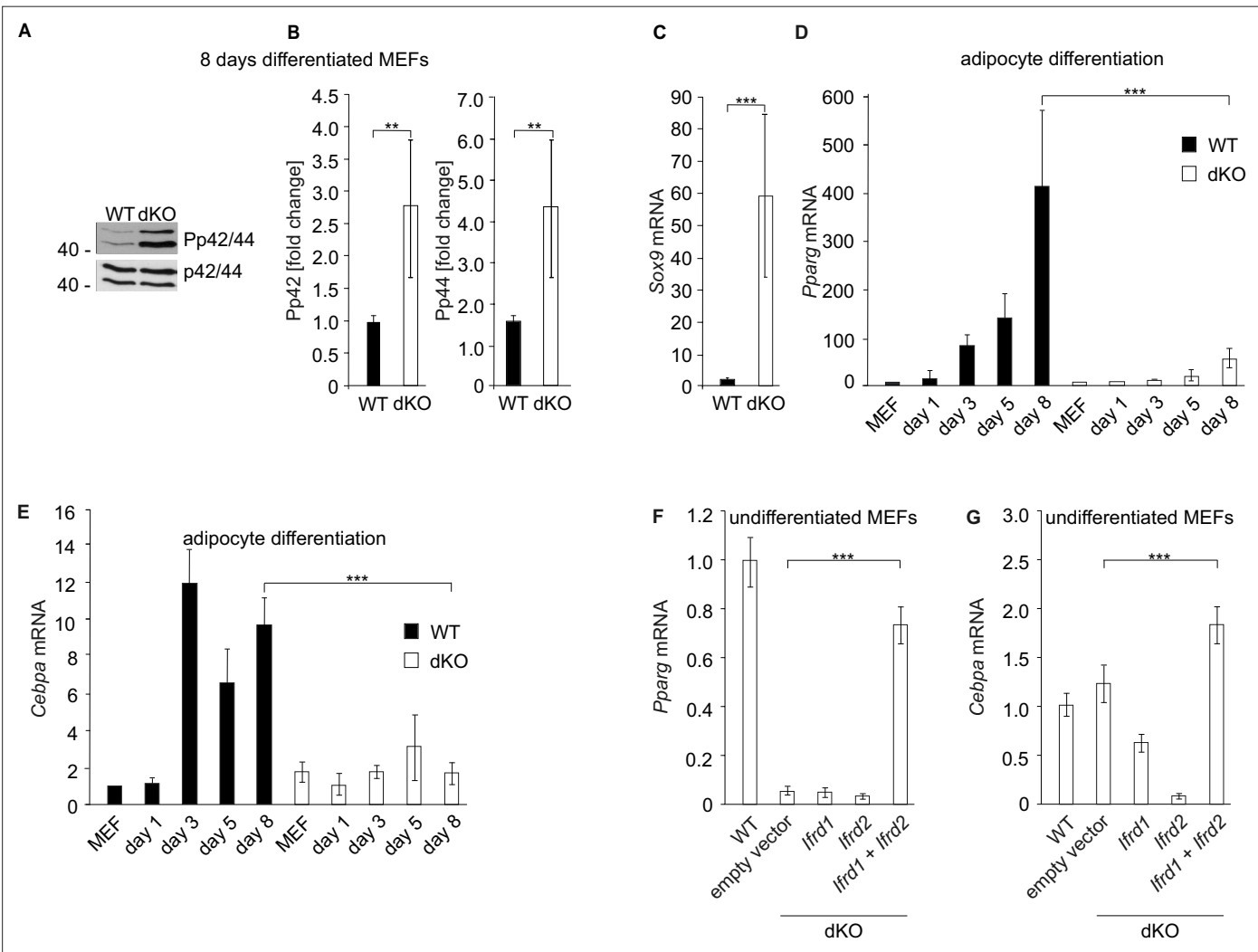

**Figure 5.** MAPK signaling and *Sox9* expression are induced while adipocyte differentiation regulatory genes are downregulated in double knockout (dKO) mouse embryonic fibroblasts (MEFs). (**A**) Representative western blots of phospho-p44, phospho-p42, p44, and p42 from 8-day adipocyte differentiation cocktail-treated MEFs. (**B**) Quantitative analysis of western blot data. Normalization on p44 and p42; n = 3. Error bars indicate standard deviations. Wild type (WT) values were set as 1; **p<0.01. (**C**) *Sox9* expression was measured by qPCR in 8-day adipocyte-differentiated WT and dKO MEFs. *Sox9* values were normalized on *GADPH* expression. WT MEF values were set as 1, n = 3. Data shown are mean ± STD. Student's *t*-test ***p<0.001. (**D**) *Pparg* and (**E**) *Cebpa* mRNA levels were downregulated in dKO adipocyte-differentiated MEFs. Gene expression was measured by qPCR during the treatment with the adipocyte differentiation cocktail. Values were normalized on *GADPH* expression. WT MEF values were set as 1, n = 3. Data shown are mean ± STD, *** p<0.001. (**F**) Ectopic co-expression of *Ifrd1* and *Ifrd2* in undifferentiated MEFs significantly increased levels of adipogenic genes *Pparg* and (**G**) *Cebpa*. Data shown are mean ± STD, Student's *t*-test ***p<0.001.

of *Ifrd1*, *Ifrd2* and their combined knockout on the *Dlk1* levels in brown adipose tissue (BAT) was analyzed. Western blot analysis identified a significant (p<0.001) increase in Dlk1 protein levels in BAT samples in knockout mice of all three genotypes (*Figure 4I*). The MEK/ERK pathway was similarly as in gonadal WAT, upregulated also in MEFs generated from dKO mice (*Figure 5A*). The phosphorylation of p42 and of p44 was upregulated 3-fold and 4.3-fold, respectively (p<0.01), in the adipocyte-differentiated dKO when compared to the WT MEFs (*Figure 5B*). Previously, activation of MEK/ERK by *Dlk1* was shown to upregulate the expression of the transcription factor *Sox9*, resulting in the inhibition of adipogenesis (*Sul, 2009*). Therefore, we measured *Sox9* mRNA expression by RT qPCR in 8-day adipocyte-differentiated MEFs. *Sox9* expression was significantly (p<0.001) upregulated in 8-day adipocyte-differentiated dKO when compared to WT MEFs (*Figure 5C*). *Pparg* and *Cebpa* mRNA levels were both continuously upregulated during the 8-day differentiation protocol of WT MEFs (*Figure 5D and E*). On the contrary, no significant increase in *Pparg* and *Cebpa* mRNA levels was found in dKO MEFs (*Figure 5D and E*). A rescue experiment showed that only the co-expression of *Ifrd1* and *Ifrd2* strongly increased the expression of the *Pparg* in undifferentiated dKO MEFs

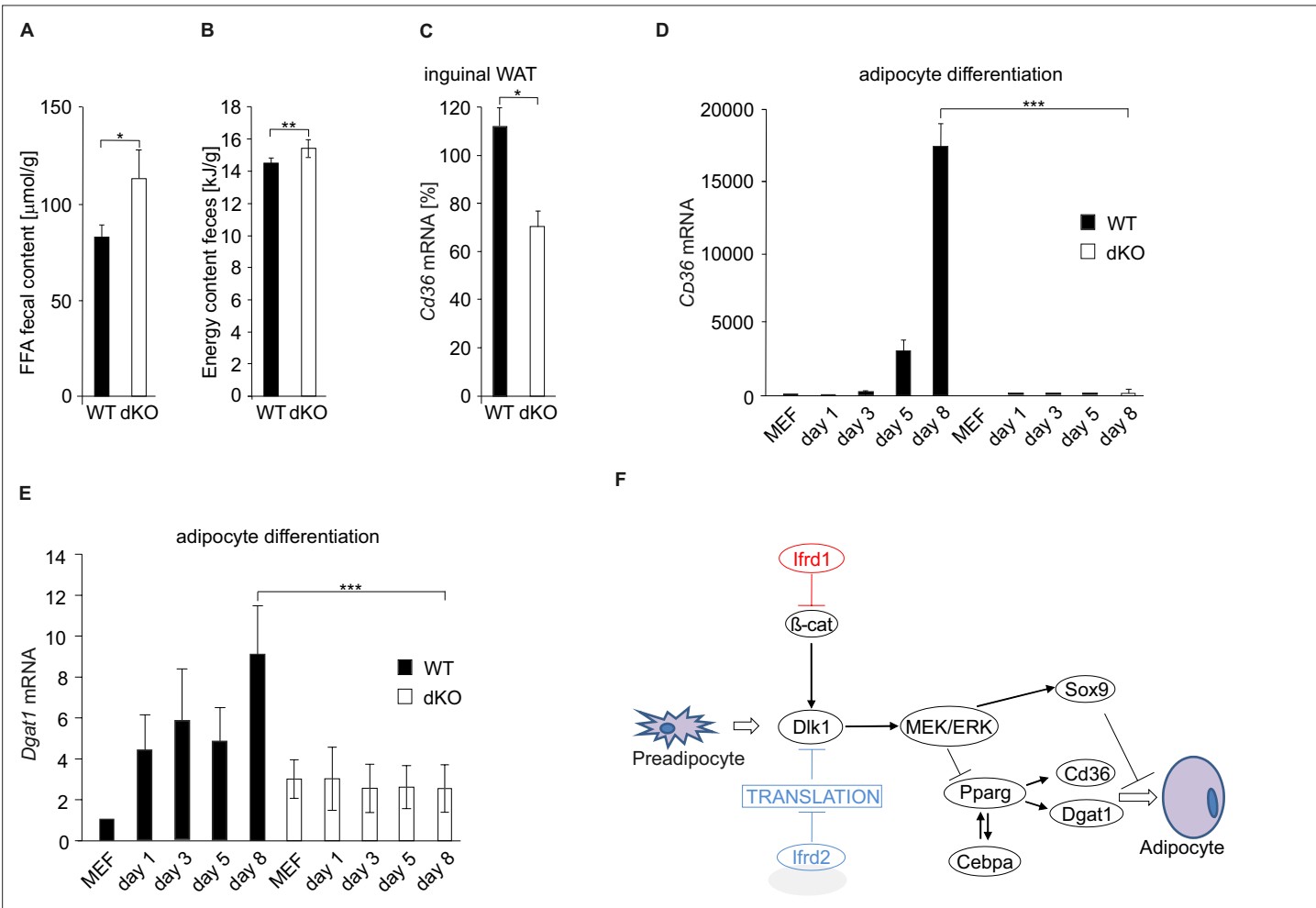

**Figure 6.** Free fatty acid (FFA) uptake is inhibited in double knockout (dKO) mice. (**A**) FFA concentrations in feces of wild type (WT) and dKO mice. Feces were collected every second day, and the composition of excreted lipids was determined by capillary gas chromatography. Data shown are mean ± STD, *p<0.05. (**B**) Energy content of dried egested feces samples was determined by bomb calorimetry. Data shown are mean ± STD, **p<0.01. (**C**) qPCR analysis of *Cd36* mRNA levels in inguinal white adipose tissue (WAT). Data shown are mean ± STD, *p<0.05. (**D**) qPCR analysis of *Cd36* in mouse embryonic fibroblasts (MEFs) differentiating into adipocytes. Relative expression levels were normalized on *GAPDH* expression. WT MEF values were set as 1. Data shown are mean ± STD, ***p<0.001. (**E**) *Dgat1* mRNA levels were downregulated in dKO adipocyte-differentiated MEFs. Data shown are mean ± STD, ***p<0.001. (**F**) Proposed model of Ifrd1 and Ifrd2 molecular mechanisms of action during adipocyte differentiation. Two parallel mechanisms leading to a deficiency in adipocyte differentiation: Ifrd1-regulated Wnt signaling affected *Dlk1* transcription and Ifrd2 acting as a translational inhibitor also contributed to the regulation of adipogenesis.

(p<0.001), almost up to the levels of WT MEFs (*Figure 5F*). The expression of *Cebpa* was also strongly upregulated by the co-expression of *Ifrd1* and *Ifrd2* (p<0.001) (*Figure 5G*). The conclusion of these results was that *Ifrd1* and *Ifrd2* regulate the expression of both *Pparg* and *Cebpa*, crucial regulators of adipocyte differentiation.

## Lipid absorption is reduced in *Ifrd1* and *Ifrd2* dKO mice

dKO mice were (*Figure 1—figure supplement 1B*, *Figure 1—figure supplement 2B*) leaner than their WT littermates despite identical food intake and RER. Therefore, the possibility that dKO mice store energy ectopically was tested. In the feces of dKO mice fed with HFD were identified significantly higher (p<0.05) amounts of free fatty acids than in that of their WT siblings (*Figure 6A*). Secondly, the energy content of dried feces from dKO mice determined by bomb calorimetry was significantly higher (p<0.001) than that of WT mice (*Figure 6B*). *Pparg* induces the expression of *Cd36*, a very long chain fatty acids (VLCFA) transporter in heart, skeletal muscle, and adipose tissues (*Coburn et al., 2000*). The regulation of *Cd36* by *Pparg* contributes to the control of blood lipids. Interestingly, *Cd36* null mice exhibit elevated circulating LCFA and TG levels consistent with the phenotype of dKO mice and *Cd36* deficiency partially protected from HFD-induced insulin resistance (*Wilson et al., 2016*). Because of downregulated *Pparg* levels in adipose tissues of dKO mice, the regulation of *Cd36* was studied more in detail as well. In inguinal WAT from dKO mice were found strongly reduced *Cd36* mRNA expression levels (64% of the WT values) (*Figure 6C*). Next, we analyzed *Cd36* in WT and dKO MEFs before onset and during adipocyte differentiation. As long as *Cd36* mRNA expression substantially increased in WT MEFs, there were almost undetectable transcript levels of *Cd36* in dKO cells treated with the adipocyte differentiation cocktail (*Figure 6D*). Diacylglycerol acyltransferase 1 (Dgat1), a protein associated with the enterocytic TG absorption and intracellular lipid processing (*Nozaki et al., 1999*), is besides Cd36 another target gene of adipogenesis master regulator *Pparg* (*Koliwad et al., 2010*). *Dgat1* mRNA levels are strongly upregulated during adipocyte differentiation (*Cases et al., 1998*), its promoter region contains a Pparg binding site (*Ludwig et al., 2002*), and *Dgat1* is also negatively regulated by the MEK/ERK pathway (*Tsai et al., 2007*). *Dgat1* expression was shown to be increased in *Ifrd1* transgenic mice (*Wang et al., 2005*), and its expression was decreased in the gut of HFD-fed *Ifrd1* KO mice (*Yu et al., 2010*). Importantly, *Dgat1* expression in adipocytes and inguinal WAT is upregulated by *Pparg* activation (*Koliwad et al., 2010*). Therefore, to analyze the role of *Ifrd1* and *Ifrd2* on the regulation of this protein involved in adipogenesis and TG processing, *Dgat1* mRNA levels were measured during the differentiation of WT and dKO MEFs into adipocytes. As long as *Dgat1* expression substantially increased during the differentiation of WT MEFs, there was no difference in *Dgat1* mRNA levels in dKO cells treated with the adipocyte differentiation cocktail (*Figure 6E*). Gene expression analyses showed that *Ifrd1* and *Ifrd2* regulated expression of multiple proteins involved both in adipocyte differentiation and in fat uptake, thereby contributing to the lean phenotype of dKO mice through multiple means.

## Discussion

Experimental data presented here show that simultaneous depletion of *Ifrd1* and of its orthologue *Ifrd2*, a protein recently identified as a translational inhibitor, caused severe reduction of adipose tissues and resistance against high fat-induced obesity in mice. Two parallel mechanisms were identified, leading to a deficiency in adipocyte differentiation. Firstly, *Ifrd1*-regulated Wnt signaling affected *Dlk1* transcription, and secondly, experimental evidence proved that Ifrd2 acted as a translational inhibitor that controls Dlk1 protein levels, thereby contributing to the regulation of adipogenesis (*Figure 6F*).

dKO mice were phenotypically similar to both *Cd36*-deficient and *Dlk1* transgenic mice, namely in decreased amounts of WAT and resistance to HFD-induced obesity. Previous studies showed that overexpression of *Ifrd1* caused increased intestinal lipid transport, resulting in elevated body weight gain during HFD feeding (*Wang et al., 2015*). Knockout of *Ifrd1* and *Ifrd2* impaired absorption of free fatty acids from the lumen into enterocytes and reduced rate of fat absorption from intestines into the circulation. Simultaneously, higher concentrations of free fatty acids in the feces of dKO mice (*Figure 6A*) suggested that mice lacking *Ifrd1* and *Ifrd2* suffer from an intestinal lipid uptake deficiency, yet another contributing reason for the lean phenotype of these mice. In dKO mice, *Pparg*

and *Cebpa* levels were inhibited, induced MEK/ERK pathway, and decreased expression of *Cd36* and *Dgat1*, all hallmarks of upregulated *Dlk1*, a known adipogenesis inhibitor. *Ifrd1* and *Ifrd2* acted after the commitment to the preadipocyte stage since the elevated *Dlk1* and β-catenin levels found in WAT of dKO mice were characteristic of the preadipocyte stage (*Gautam et al., 2017*).

Disruption of Wnt signaling in embryonic fibroblasts results in spontaneous adipocyte differentiation (*Bennett et al., 2003*), and in contrast, stabilized β-catenin keeps cells in the preadipocyte stage (*Ross et al., 2000*). A subpopulation of the SVF of adipose tissue is adipogenic and at the same time has a weaker Wnt/β-catenin signal (*Hu et al., 2015*). Consistent with this knowledge, upregulated β-catenin protein levels and increased Wnt signaling activity in WAT of *Ifrd1* single and dKO MEFs were found. It has to be mentioned that the data presented here differ from those published by Nakamura et al., who showed that following the *Ifrd1* overexpression in adipocytes Wnt/β-catenin signaling was upregulated and inhibited oil red O staining (*Nakamura et al., 2013*). However, one has to take into consideration the difference in cell systems used in these two studies. In contrast to Nakamura's study where results were obtained in 3T3-L1 cells fibroblasts overexpressing *Ifrd1*, the data presented here document the role of endogenous *Ifrd1* in cells derived from WT or knockout mice. Moreover, *Ifrd1* was found to be ubiquitously expressed in all WT mouse organs without any pretreatment such as hypoxia. On the other hand, Nakamura et al. identified upregulated *Ifrd1* expression levels in WAT of obesity model mice. This result, however, fully supports the findings of lean phenotype in *Ifrd1* and *Ifrd2* dKO mice.

In dKO MEFs, induced for adipocyte differentiation, the increase in Wnt signaling led to upregulated binding of β-catenin to the *Dlk1* gene regulatory elements, resulting in sustained *Dlk1* expression. In adipogenesis, *Dlk1* expression is downregulated through histone methylation by Coprs and PRMT5 proteins that prevent β-catenin binding to the *Dlk1* gene (*Paul et al., 2015*). Interestingly, in dKO MEFs weaker binding of dimethylated H4R3 to the *Dlk1* gene was found, consistent with the previous findings on the negative role of *Ifrd1* in epigenetic regulation of gene expression including the PRMT5 activity (*Lammirato et al., 2016*).

The difference in the effects of ectopic expression of *Ifrd1*, *Ifrd2* and their co-expression on *Dlk1* levels (*Figure 3B*) confirmed the hypothesis that proteins Ifrd1 and Ifrd2 regulate *Dlk1* levels via two independent pathways/mechanisms. Despite the fact that *Ifrd2* knockout had no effect on Wnt signaling, it nevertheless affected *Dlk1* levels, suggesting a contribution of Ifrd2. Since protein Ifrd2 was identified as a novel factor translationally inactivating ribosomes (*Brown et al., 2018*), downregulation of the protein synthesis machinery is an essential regulatory event during early adipogenic differentiation (*Marcon et al., 2017*). Therefore, there was a possibility that Ifrd2 regulates Dlk1 protein levels through translational regulation. The experimental data presented here confirmed specific regulation of *Dlk1* through this mechanism. Elevated *Dlk1* levels in dKO MEFs activated the MEK/ERK pathway, thereby decreasing *Pparg* and *Cebpa* levels important for adipogenic differentiation. Besides, the expression of *Sox9* and *Hes1* was upregulated, suggesting that the dKO MEFs keep their proliferative state and cannot enter differentiation into mature adipocytes (*Kim et al., 2007*).

Downregulation of *Pparg* and *Cebpa* implied changes in the expression of downstream adipogenesis-related genes. Among them, decreased *Cd36* and *Dgat1* levels were a plausible explanation of the dKO mice lean phenotype since it is known that both proteins play a functional role in the differentiation of murine adipocytes and their deficiency impairs fat pad formation independent of lipid uptake (*Christiaens et al., 2012*). The results presented here implicate that Ifrd1 and Ifrd2 play a up to now unknown role in the regulation of Cd36 and therefore possibly contribute to Cd36-related pathogenesis of human metabolic diseases, such as hypoglycemia (*Nagasaka et al., 2011*), hypertriglyceridemia (*Kashiwagi et al., 2001*), and disordered fatty acid metabolism (*Glazier et al., 2002*; *Tanaka et al., 2001*).

It is possible that due to the reduced *Cd36* expression levels in intestines, *Ifrd1 Ifrd2* dKO mice accumulated lower amounts of body fat and were resistant to diet-induced obesity also because of limited intestinal fat uptake. Furthermore, *Ifrd1 Ifrd2* dKO mice displayed increased plasma TG levels due to impaired clearance of VLCFA transport by skeletal muscles. Co-expression of *Ifrd1* and *Ifrd2* in dKO myoblasts partially rescued the Cd36 expression on the transcriptional level as documented by the increase of the *Cd36* promoter activity. Rescuing the free fatty acids transport by *Cd36* overexpression in dKO myoblasts confirmed the hypothesis that the deficit in fatty acid transport was due to

the diminished *Cd36* expression. Thus, *Ifrd1 Ifrd2* dKO mice accumulated lower amounts of adipose tissue also because of impaired lipid transport.

It was previously shown that *Ifrd1* is involved in both chow and HFD conditions in intestinal TG absorption. The experimental data shown here document for the first time the role of *Ifrd1* and its orthologue *Ifrd2* in adipocyte differentiation. Moreover, they confirm the physiological function of *Ifrd2* as a translational inhibitor. Surprisingly, although *Ifrd1* and *Ifrd2* share sequence homology and a functional role in the adipogenesis, they use two independent regulatory mechanisms.

## Methods

### Generation of animal models

All animal experiments were performed in accordance with Austrian legislation BGB1 Nr. 501/1988 i.d.F. 162/2005.

Mice lacking *Ifrd1* were described previously (*Vadivelu et al., 2004*). *Ifrd2* KO mice generation: targeting construct contained *Ifrd2* gene locus exons. A loxP site and a neomycin resistance gene were inserted at position 105515 (AY162905). The neomycin cassette was flanked by two frt sites. A second loxP site was inserted at position 102082 (AY162905). Further downstream, 15 additional nucleotides, part of intron 1, exon 2 (splice site: donor and acceptor), and the CDS of hrGFP from the Vitality hrGFP mammalian expression vector pIRES-hrGFP-2a (Stratagene) were added. The targeted construct was electroporated into Sv129 mouse ES cells. After the selection with G418, single-cell clones were screened by PCR and confirmed by Southern blot analysis. After Cre recombinase treatment, cell clones were screened by PCR. Clones with floxed gene deleted were used for blastocyst injection into C57BL/6J mice. Two male chimeras with ≥40% or more agouti coat color were mated to C57BL/6J females. The knockout mouse strain was derived from one male mouse carrying the allele of interest. Heterozygous mice were back-crossed to C57BL6/J mice for nine generations. dKO mice were generated by crossing the *Ifrd2* KO mice with the *Ifrd1* single KO mice. The resulting mice were screened by PCR and double heterozygous mice used for further breeding until homozygosity. In order to achieve maximal homogeneity of experimental groups, in all experiments presented here we used only male mice.

### *Ifrd2* knockout Southern blot analysis

XbaI restriction sites (position 109042; position 102079; position 92253) were located in the *Ifrd2* locus (AY162905). Fragment detected by the Southern probe: 9.6 kb wt (*Figure 1—figure supplement 2F*, inset, band V); 15 kb in deleted locus (band IV). The 566-bp-long probe for hybridization was from genomic DNA (96605–97171; AY162905). PCR primer sequences: RK 150 Fwd: 5′-GGTCCTGCCACTAATGCACTG-3′; RK 151 Rev: 5′-GCAGACAGATGCCAGGAAGAC-3′.

### *Ifrd2* knockout PCR genotyping analysis

hrGFP insert in the 3′ UTR was detected by primer GFP2 5′-AGCCATACCACATTTGTAGAG-3′ and RK101 3′ UTR (5′-TGATGATAGCTTCAAAGAGAA-3′; 100617–100591 of the *Ifrd2* locus (AY162905). PCR product 1700 bp. *Ifrd2* detection: RG1; 5′-TGTGGCCTTTATCCTGAGTC-3′; 102286–102266) and RG2; 5′-TGGCTTCATTTACACTACTCCTT-3′; 101860–101882 primers (*Figure 1—figure supplement 2F*). WT allele PCR product 426 bp and the targeted allele PCR product 1772 bp (*Figure 1—figure supplement 2G*). *Ifrd1* genotype was tested as explained previously (*Vadivelu et al., 2004*; *Figure 1—figure supplement 2H*).

### Growth monitoring and body composition measurement

Mice were weaned at 3 wk, regular chow diet, and weighed weekly. DEXA was measured with Norland scanner (Fisher Biomedical). Micro-CT experiments were performed using vivaCT 40 (Scanco Medical AG). The scans were performed using 250 projections with 1024 samples, resulting in a 38 µm isotropic resolution. Tube settings: 45 kV voltage, 177 µA current, integration time 300 ms per projection. Image matrix 1024 * 1024 voxels and a grayscale depth of 16 bit. The length of the image stack was individually dependent, starting from the cranial end of the fist lumbar vertebrae to the caudal end of the fifth lumbar vertebrae. The image reconstruction and postprocessing were performed using the Scanco Medical system software V6.6. For the adipose tissue evaluation, an IPL (image processing

language) script by Judex et al., provided by Scanco Medical AG, was modified to the scanner individual parameters, leading to two values lower threshold than in the original script for the adipose tissue filters 76. The script calculated the total abdominal volume without potential air in the cavities. A separation of subcutaneous and visceral fat mass was used only for visualization. For the quantitative fat mass analysis, we used 56 male 5–12-month-old mice (mean age 9.35 ± 2.03 mo) reflecting an adult to mid-aged cohort defined by *Flurkey et al., 2007*; *Neeland et al., 2013* #4207. The development of adipose tissue in healthy mice is stable between 4 and 12 mo (*Lemonnier, 1972*). Therefore, sex and age could have only very limited influence on the experimental results. For the quantitative comparison, the percent contribution of the abdominal fat to the body weight was calculated using a mean weight of 0.9196 g/ml for adipose tissue (*Neeland et al., 2013*). Statistics were performed using an ANCOVA with a Bonferroni corrected post hoc testing on the µCT fat data and the body mass data.

## Metabolic measurements

The indirect calorimetry trial monitoring gas exchange, activity, and food intake was conducted over 21 hr (PhenoMaster TSE Systems). Body mass and rectal body temperature before and after the trial were measured. The genotype effects were statistically analyzed using one-way ANOVA. Food intake and energy expenditure were also analyzed using a linear model including body mass as a co-variate.

## HFD feeding

Age-matched (7-week-old) male *Ifrd1-/- Ifrd2-/-* and WT mice were caged individually and maintained up to 3 wk on a synthetic, HFD diet (TD.88137; Ssniff). Animals were weighted every fourth day between 08:00 and 10:00. Intestines, liver, muscles, and adipose tissue were collected for total RNA and protein isolation. Unfixed intestines were flushed with PBS using a syringe, embedded in Tissue-Tek (Sakura, 4583) and frozen in liquid nitrogen for immunohistochemical analysis.

## Quantitative food consumption and fecal fat determination

Adult mice were acclimatized to individual caging and to the HFD for a week, and monitored for weight gain and their food intake daily. The daily food intake data were pooled for the following 7 d, and the food intake was estimated (g/day). Feces were collected daily and weighted for 7 d after the second week of the HFD consumption. Lipids were extracted and methylated according to *Lepage et al., 1989*. After freeze-drying and mechanical homogenization, aliquots of feces were subjected to the same procedure as described by *Lepage et al., 1989*. The resulting fatty acid methyl esters were analyzed by gas chromatography to measure the total and individual amounts of major fatty acids (*Minich et al., 2000*). The energy content of dried egested feces samples (~1 g per sample) was determined by bomb calorimetry (IKA C 7000, IKA, Staufen, Germany) (*Pfluger et al., 2015*).

## Plasma cholesterol and triglyceride analyses

Serum cholesterol and TG were measured using cholesterol/TG reagent (Cobas, Roche) according to the manufacturer's instructions. Lipoprotein profiles were analyzed by fractionation of pooled serum using two Superose 6-columns (Cytiva) in series (FPLC), followed by cholesterol measurement (*Demetz et al., 2020*).

## Antibodies, viral and cDNA constructs

Antibodies: anti-Cd36 AbD Serotec (MCA2748), Abcam (ab36977), p44/42 MAPK (Erk1/2), and Phospho-p44/42 MAPK (Erk1/2) (Thr202/Tyr204) Cell Signaling Technology (9102, 9101), β-catenin antibody Sigma (C2206), anti-Dlk1 Abcam (ab119930), anti-histone H4R3me2s antibody Active Motif (61187), and anti-Ifrd1 (Sigma-Aldrich Cat# T2576, RRID:AB_477566). For ChIP experiments, anti-Ifrd1 (*Vietor et al., 2002*) and anti- Ifrd2 (*Vadivelu et al., 2004*) rabbit polyclonal antibodies previously proven for ChIP suitability in *Lammirato et al., 2016* were used. pTOPflash reporter construct was a gift from H. Clevers (University of Utrecht, Holland). *Ifrd1* construct was described previously (*Vietor et al., 2002*). Partial CDS of *mIfrd2* was amplified by PCR and cloned into pcDNA3.1(-)/MycHis$_6$ (Invitrogen).

## Cell culture and adipocyte differentiation

MEFs were generated from 16-day-old embryos. After dissection of head for genotyping and removal of limbs, liver and visceral organs, embryos were minced and incubated in 1 mg/ml collagenase

(Sigma-Aldrich, C2674) 30 min at 37°C. Embryonic fibroblasts were maintained in growth medium containing DMEM high glucose (4.5 g/l), sodium pyruvate, L-glutamine, 10% FCS (Invitrogen, 41966029), and 10% penicillin/streptomycin (Sigma-Aldrich, P0781) at 37°C in 5% $CO_2$. Adipocyte differentiation treatment was medium with 0.5 mM 3-isobutyl-1-methylxanthine (Sigma-Aldrich, I5879), 1 µM dexamethasone (Sigma-Aldrich, D4902), 5 µg/ml insulin (Sigma-Aldrich, I2643), and 1 µM tosiglitazone (Sigma-Aldrich, R2408). After 3-day growth medium containing 1 µg/ml insulin, cells were differentiated for 8 d. To visualize lipid accumulation, adipocytes were washed with PBS, fixed with 6% formaldehyde overnight, incubated with 60% isopropanol, air-dried, and then incubated with oil red O. Microscopic analysis was followed by isopropanol elution and absorbance measurement at 490 nm. MEF's genotype was controlled by PCR reactions. Stromal vascular cells were prepared exactly according to the previously published protocol (*Aune et al., 2013*). A possible mycoplasma contamination was routinely controlled by PCR. All cells used for experiments were mycoplasma negative.

## Construction of expression plasmids and generation of stable cell lines

The pRRL CMV GFP Sin-18 plasmid (*Zufferey et al., 1998*) was used to generate *Ifrd1*, *Ifrd2*, and *Dlk1* expression of lentiviral constructs. For this, the corresponding cDNAs were cloned into the BamHI and SaII sites of the pRRL CMV GFP Sin-18 plasmid. A cap-independent translation enhancer (CITE) fused to the puromycin resistance pac gene and the woodchuck hepatitis virus post-transcriptional regulatory element (WPRE) were introduced downstream of the *Ifrd1*, *Ifrd2*, and *Dlk1* coding sequences. All DNA constructs were verified by sequencing.

To generate the *Dlk1* shRNA lentiviral vectors, oligonucleotides targeting either all *Dlk1* mRNA splice variants (*Dlk1* total: 5′-GATCCCCAGATCGTAGCCGCAACCAATTCAAGAGATTGGTTGCGGCT ACGATCTTTTTTGGAAA-3′) or only *Dlk1* mRNA splice variants containing the extracellular cleavage sequence (*Dlk1*PS: 5′-GATCCCCTCCTGAAGGTGTCCATGAATTCAAGAGATTCATGGACACCTTCA GGATTTTTGGAAA-3′) (*Mortensen et al., 2012*) were fused with the H1 promoter and cloned into the pRDI292 vector as reported (*Reintjes et al., 2016*). The GFP-targeting pRDI-shRNA-GFP plasmid was used as a control (*Reintjes et al., 2016*).

The viral supernatants were obtained as described previously (*Leitner et al., 2022*), concentrated with Retro-X Concentrator (Clontech, Takara Bio), and used to infect WT, *Ifrd1*, and *Ifrd2* single and dKO MEFs. The selection was carried out for 2 wk in DMEM supplemented with 10% (v/v) FBS, 100 U/ ml penicillin and 100 µg/ml streptomycin, and 2 µg/ml puromycin (Sigma-Aldrich, P7255).

## Polysome profiling

Polysome profiling was performed as described in *Savant-Bhonsale and Cleveland, 1992*, with modifications. The day before harvesting the cells, continuous 15–45% (w/v) sucrose gradients were prepared in SW41 tubes (Beckman) in polysome gradient buffer (10 mM HEPES-KOH, pH 7.6; 100 mM KCl; 5 mM $MgCl_2$) employing the Gradient Master ip (Biocomp) and stored overnight at 4°C. All steps of the protein extraction were performed on ice. Exponentially growing cells were washed twice with ice-cold DPBS (Gibco) supplemented with 100 µg/ml f. c. cycloheximide, scraped in 300 µl polysome lysis buffer (10 mM HEPES-KOH, pH 7.6; 100 mM KCl; 5 mM $MgCl_2$; 0.5% IGEPAL CA-630; 100 µg/ ml cycloheximide) supplemented with 0.1 U/µl murine RNase Inhibitor (NEB), and passed through a G25 needle 25 times. Nuclei were pelleted at 16,000 × *g* for 6 min at 4°C, and the supernatants were carefully layered onto the sucrose gradients. Samples were centrifuged at 35,000 rpm for 2 hr at 4°C (with brakes switched off) using an SW 41 Ti rotor (Beckmann). Twenty fractions of 0.6 ml were collected using a peristaltic pump P1 (Amersham Biosciences), and polysome profiles were generated by optical density measurement at 254 nm using optical unit UV-1 (Amersham Biosciences) and chart recorder Rec 111 (Amersham Biosciences).

## Analysis of translation by metabolic labeling

Pulse labeling of proteins was performed as described before (*Popow et al., 2015*), with the following changes: $1 × 10^6$ wild-type and dKO MEFs cells per plate were seeded and cultivated for 24 hr. Cells were washed twice with PBS followed by one wash with labeling medium (ESC medium without cysteine and methionine) and a 30 min incubation in labeling medium. To label newly translated products, 200 µCi of [$^{35}$S]-methionine (10 mCi/ml; Hartman Analytic) were added and the cells were

incubated for 1 hr. Cells were washed and incubated another 10 min in standard medium at 37°C. Cells were then harvested by trypsinizing, washed once with ice-cold PBS, extracted with ice-cold RIPA buffer containing protease-inhibitors, and briefly sonicated. Proteins were fractionated by gel electrophoresis in 16% Tricine gels (Thermo Fisher, EC66955BOX) and stained with Coomassie brilliant blue, and radioactive signals were visualized by phosphorimaging. Signal intensities were quantified using the Image Studio Lite (v5.2) software.

## RT-PCR

Tissues from animals fed for 3 wk with HFD were snap-frozen and stored at –80°C. Total RNA was isolated using the TRIzol reagent (Invitrogen, 15596026). RNA was then chloroform-extracted and precipitated with isopropanol. The yield and purity of RNA were determined by spectroscopic analysis; RNA was stored at –80°C until use.

## Quantitative RT-PCR and statistics

Total RNA were treated with DNAse1 and reverse transcribed to cDNA by Revert Aid First Strand cDNA Synthesis Kit (Thermo Scientific, K1622) with oligo dT primers. Quantitative RT-PCR was performed using TaqMan probes and primer sets (Applied Biosystems) specific for CD36 (assay ID Mm00432398_m1), Dgat1 (Mm00515643_m1), Pparg (Mm00440940_m1), Cebpa (Mm00514283_s1), *Dlk1* (Mm00494477_m1), and Sox9 (Mm00448840_m1). Ribosomal protein 20 (assay ID Mm02342828_g1) was used as normalization control for quantification by the ddCt method. PCR reactions were performed using 10 µl cDNA in PikoReal 96 real-time PCR system (Thermo Scientific). Quantification data were analyzed by two-tailed, homoscedastic *t*-tests based on the assumption that variances between the two sample data ranges are equal to type 2 Student's *t*-test.

## Transient transfections and luciferase assay

pGL2-Basic (Promega, E1641) or pGLCD36 (*Shore et al., 2002*) were used as reporter constructs. Expression constructs or empty vector DNA as a control were co-transfected. pCMV-β-Gal plasmid was used to normalize for transfection efficiency. For luciferase reporter assays, $1.5 \times 10^5$ cells were seeded into 24-well plates and transfected after 24 hr with the indicated plasmid combinations using Lipofectamine Plus Reagent (Invitrogen, 15338030). The total amount of transfected DNA (2 µg DNA per well) was equalized by addition of empty vector DNA. Cells were harvested 48 hr post-transfection in 0.25 M Tris, pH 7.5, 1% Triton X-100 buffer and assayed for both luciferase and β-galactosidase activities. Luciferase activity and β-galactosidase activity were assayed in parallel using the Lucy 2 detection system (Anthos). Transfections were performed in triplicates, and all experiments were repeated several times.

## Chromatin immunoprecipitation (ChIP)

Chromatin was isolated from *Ifrd1* WT and dKO formaldehyde-treated, 8-day adipocyte-differentiated MEFs using the EpiSeeker Chromatin Extraction Kit (Abcam, ab117152). ChIP analyses were carried out as described previously (*Reintjes et al., 2016*). The sequence of the oligonucleotides for two regions of the *Dlk1* promoter, encompassing TCF and β-catenin binding sites, were as defined in *Paul et al., 2015*. Sonicated chromatin was centrifuged at 15.000 × *g* for 10 min at 4°C, and the supernatant (65 µg of sheared DNA per each IP) was diluted tenfold with cold ChIP dilution buffer containing 16.7 mM Tris-HCl pH 8.1, 167 mM NaCl, 0.01% (w/v) SDS, 1.1% (w/v) Triton X-100, and 1.2 mM EDTA with protease inhibitors. Samples were pre-cleared for 1 hr with protein A Sepharose CL-4B (Sigma-Aldrich, 17-0780-01) beads blocked with 0.2 µg/µl sonicated herring sperm DNA (Thermo Fisher, 15634017) and 0.5 µg/µl BSA (NEB, B9000S). Immunoprecipitations were performed at 4°C overnight. Immune complexes were collected with protein A Sepharose for 1 hr at 4°C followed by centrifugation at 1000 rpm and 4°C for 5 min. Beads were washed with 1 ml low salt wash buffer (20 mM Tris-HCl pH 8.1, 150 mM NaCl, 0.1% [w/v] SDS, 1% [w/v] Triton X-100 [Merck], 2 mM EDTA), high salt wash buffer (20 mM Tris-HCl pH 8.1, 500 mM NaCl, 0.1% [w/v] SDS, 1% [w/v] Triton X-100, 2 mM EDTA), LiCl wash buffer (10 mM Tris-HCl pH 8.1, 250 mM LiCl, 1% [w/v] sodium deoxycholate, 1% [w/v] IGEPAL-CA630, 1 mM EDTA) for 5 min at 4°C on a rotating wheel, and twice with 1 ml TE buffer (10 mM Tris-HCl pH 8.0, 1 mM EDTA). Protein-DNA complexes were eluted from antibodies by adding a freshly prepared elution buffer containing 1% SDS and 0.1 M NaHCO₃. The eluate was reverse cross-linked by adding

NaCl to a final concentration of 0.2 M and incubating at 65°C for 4 hr. Afterward the eluate was treated with proteinase K at 45°C for 1 hr. The immunoprecipitated DNA was then isolated by phenol/chloroform precipitation and used as a template for real-time quantitative PCR. The primer pairs specific for regulatory regions of the *Dlk1* gene were selected as described before (*Paul et al., 2015*). Reactions with rabbit IgG or with 1.23% of total chromatin (input) were used as controls. For real-time quantitative PCR, a PikoReal System was used. Signals were normalized to input chromatin and shown as % input. The raw cycle threshold (Ct) values of the input were adjusted to 100% by calculating raw Ct – log2(100/input). To calculate the % input of the immunoprecipitations, the equation 100 × 2[Ct (adjusted input to 100%) – Ct (IP)] was applied.

## Statistical analyses

Statistical analyses were performed with one-way ANOVA, Student's unpaired *t*-test using GraphPad Prism version 9.2 (GraphPad, La Jolla, CA) software, or as indicated in the legends. p-Value is indicated by asterisks in the figurees: *p≤0.05, **p<0.01, ***p<0.001, ****p<0.0001. Data from SVF cells were analyzed using ordinary one-way ANOVA with Holm–Šidák's multiple-comparisons test.

## Acknowledgements

We thank Robert Kurzbauer for generation of dKO mice, Stephan Geley, Laura M de Smalen, Laura De Gaetano, and Karin Schluifer for the technical assistance, Christiane Heim for serum analyses and free fatty acid uptake measurements, Frans Stellaard for the analysis of fatty acids content in feces, Mayra Eduardoff for RNA processing for Affymetrix chip analysis, Alexander Magnutzki for advice with the statistical analyses of data, and David Teis and Zlatko Trajanoski for critical reading of the manuscript. Furthermore, we would like to thank Dr Paul Shore for providing us with the pGLCD36 construct. We are indebted to the staff at the Animal Facility of Innsbruck Medical University for their care of our mice. This work was supported by P18531-B12 and P22350-B12 grants from the Austrian FWF grant agency to Ilja Vietor and by the German Federal Ministry of Education and Research (Infrafrontier grant 01KX1012) to Martin Hrabe de Angelis.

## Additional information

### Funding

| Funder | Grant reference number | Author |
| --- | --- | --- |
| Austrian Science Fund | P18531-B12 | Ilja Vietor |
| Austrian Science Fund | P22350-B12 | Ilja Vietor |
| Helmholtz Zentrum München | 01KX1012 | Martin Hrabe de Angelis |

The funders had no role in study design, data collection and interpretation, or the decision to submit the work for publication.

### Author contributions

Ilja Vietor, Conceptualization, Resources, Data curation, Formal analysis, Supervision, Funding acquisition, Validation, Investigation, Visualization, Methodology, Writing – original draft, Project administration, Writing – review and editing; Domagoj Cikes, Conceptualization, Formal analysis, Investigation, Writing – original draft; Kati Piironen, Theodora Vasakou, Formal analysis, Investigation; David Heimdörfer, Philipp Eller, Egon Demetz, Investigation; Ronald Gstir, Investigation, Methodology; Matthias David Erlacher, Resources, Validation; Ivan Tancevski, Conceptualization, Resources, Validation, Investigation; Michael W Hess, Conceptualization, Validation; Volker Kuhn, Resources, Formal analysis, Validation; Gerald Degenhart, Resources, Data curation, Formal analysis, Validation, Investigation; Jan Rozman, Conceptualization, Investigation; Martin Klingenspor, Writing – review and editing; Martin Hrabe de Angelis, Funding acquisition; Taras Valovka, Conceptualization, Data curation, Formal analysis, Validation, Investigation, Visualization, Writing – review and editing; Lukas A Huber, Conceptualization, Formal analysis, Funding acquisition, Validation, Visualization, Writing – review and editing

## Author ORCIDs

Ilja Vietor https://orcid.org/0000-0002-1391-6793
Domagoj Cikes https://orcid.org/0000-0003-0350-5672
Gerald Degenhart http://orcid.org/0000-0002-9961-1084
Jan Rozman https://orcid.org/0000-0002-8035-8904
Martin Klingenspor https://orcid.org/0000-0002-4502-6664
Martin Hrabe de Angelis https://orcid.org/0000-0002-7898-2353
Lukas A Huber https://orcid.org/0000-0003-1116-2120

## Ethics

All animal experiments were performed in accordance with Austrian legislation BGB1 Nr. 501/1988 i.d.F. 162/2005.

## Decision letter and Author response

Decision letter https://doi.org/10.7554/eLife.88350.sa1
Author response https://doi.org/10.7554/eLife.88350.sa2

---

# Additional files

## Supplementary files

• MDAR checklist

## Data availability

All data generated or analyzed during this study are included in the manuscript, figures and associated source data files.

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
