## [Editor Report]

This study provides important new insights into the molecular regulation of adipocyte differentiation. Two molecules, TIS7 and SKMc15, are shown to regulate the activity of the key transcriptional regulator DLK-1 via discrete mechanisms – one involving transcription and the other translation. These findings add valuable information to the well known roles of Wnt/catenin and PPARg on adipocyte differentiation and will provide an advance for those interested in the role of adipocytes in whole body metabolism.

---

## [Decision Letter]

[Editors' note: this paper was reviewed by Review Commons.]

Thank you for submitting your article "The negative adipogenesis regulator DLK1 is transcriptionally regulated by TIS7 (IFRD1) and translationally by its orthologue SKMc15 (IFRD2)" for consideration by *eLife*. Your article has been reviewed by 3 peer reviewers at Review Commons, and the evaluation at *eLife* has been overseen by a Reviewing Editor and David James as the Senior Editor.

Based on the previous reviews and the revisions, the manuscript has been improved but there are some remaining issues that need to be addressed, as outlined below:

This study provides new insights into the molecular regulation of adipocyte differentiation. Two molecules, TIS7 and SKMc15, are shown to regulate the activity of the key transcriptional regulator DLK-1 via discrete mechanisms – one involving transcription and the other translation. These findings add additional information to the well known roles of Wnt/catenin and PPARg on adipocyte differentiation.

The authors have extensively addressed the comments of the referees and all referees are convinced that the manuscript is now improved and the mechanistic details of the link between SKMc15 and regulation of adipogenesis is now strengthened. However, as described below there are still some important issues that need to be addressed by the authors prior to publication. Many of these require editorial changes but there are some minor experimental details that need to be addressed. Also essential is that you discuss more thoroughly the lipid absorption issue as a contributor to the dKO mouse phenotype.

Specific Issues

1) All energy balance measurements need to be included in the manuscript, not just shown to reviewers. Readers will want to see them.

2) It is not quite clear how food intake is expressed. Since the mice are significantly smaller, it might be more appropriate to express the data as g of food/g of mouse, as a smaller mouse is likely to eat less.

3) In complementation experiments, it would be useful to know the levels of overexpression.

4) What is the levels of blood lipids in the dKO mice, since this is the first report of their existence, and presumably the defect in intestinal lipid absorption may affect these?

5) The methods need to be carefully edited. For instance, lines 564-65 state "Small intestines were harvested for oil red O staining to detect lipid accumulation"; these data are not shown anywhere in the paper. Similarly, the section on fecal fat determination describes a protocol to analyze neutral sterols and bile acids (nowhere in the paper), but it does not describe how the free fatty acid levels were determined. One more example, line 651 alludes to "pulse labeling of mitochondrial proteins" which is not what is presented in the paper.

6) The main text should indicate at the outset that only male mice were analyzed.

7) It is incorrect to state that the dKO mice are not smaller when the graph showing that data (panel A in Figure EV1) shows a significant difference.

---

## [Author Response]

Reviewer #1 (Evidence, reproducibility and clarity (Required)):This study by Viedor et al. examines the role of TIS7 (IFRD1) and its ortholog SKMc15 (IFRD2) in the regulation of adipogenesis via their ability to modulate the levels of DLK1 (Pref-1), a well-known inhibitor of adipogenesis. They generate SKMc15 KO mice and cross them to previously published TIS7 KO mice. All 3 mutant strains show decreased fat mass, with the effect being most pronounced in double KO mice (dKO). Using mouse embryonic fibroblasts (MEFs) from mutant mice, they authors ascribe a defect in adipogenic differentiation of mutant cells to an upregulation of DLK-1. In the case of TIS7, they propose that this is due to its known inhibition of Wnt signaling, which regulates DLK-1 expression. In the case of SKMc15, they suggest a new mechanism linked to its ability to suppress translation. Overall, the work is of interest, with the finding, that SKMc15 regulates adipocyte differentiation being its novelty, and generally well done, though multiple aspects need to be improved to bolster the conclusions put forth.Major concerns:1) The main mechanism put forth by the authors to explain the inability of dKO cells to differentiate into adipocytes is the upregulation of DLK-1 levels. However, this notion is never directly tested. Authors should test if knockdown of DLK-1 in dKO cells is sufficient to correct the defect in differentiation, or if additional factors are involved.

In response to the reviewer’s concerns, we have generated two stable cell lines expressing short hairpin RNAs directed against DLK1 in the TIS7 SKMc15 dKO MEFs. With these two and the parental dKO MEF cell line, we have performed adipogenesis differentiation experiments as explained in the manuscript before. Figure EV2C (left and right panels) shows that knockdown of DLK1 with two different DLK1 shRNA constructs (targeting DLK1 with or without the extracellular cleavage site) significantly (P<0.001) increased the adipocyte differentiation ability of dKO MEFs. This result indicated that DLK1 knockdown alone was sufficient to correct the differentiation deficit of these cells.

2) There are multiple instances were the authors refer to "data not shown", such as when discussing the body length of dKO mice. Please show the data in all cases (Supplementary Info is fine) or remove any discussion of data that is not shown and cannot be evaluated.

Following three results were in the initial version of our manuscript mentioned as “data not shown”:

line 137: “body length, including the tail did not significantly differ between WT and dKO mice”line 307: “higher concentrations of free fatty acids in the feces of dKO mice”line 331: “effects of ectopic expression of TIS7, SKMc15 and their co-expression on DLK-1 levels”

In the current version of the manuscript, we provide these results as:

3) Indirect calorimetry data shown in Figure S1 should include an entire 24 hr cycle and plots of VO2, activity and other measured parameters shown (only RER and food intake are shown), not just alluded to in the legend.

Based on the reviewer’s suggestion, we present here a table (Table 1) containing all parameters measured in the indirect calorimetry experiment.

Metabolic phenotyping presented in Figure EV1B containing 21 hours measurement was performed exactly according to the standardized protocol previously published by Rozman J. *et al.* [1]. All phenotyping tests were performed following the International Mouse Phenotyping Resource of Standardized Screens (IMPReSS) pipeline routines.

4) It is surprising that the dKO mice weight so much less than WT even though their food consumption and activity levels are similar, and their RER does not indicate a switch in fuel preference. An explanation could be altered lipid absorption. The authors indicate that feces were collected. An analysis of fat content in feces (NEFAs, TG) needs to be performed to examine this possibility. The discussion alludes to it, but no data is shown.

We thank the reviewer for bringing up this important point that prompted us to present data clarifying this aspect of the metabolic phenotype of dKO mice. As shown in Figures 6A,B, while fed with HFD, dKO mice had higher concentrations of free fatty acids in the feces (109 ± 10.4 µmol/g) when compared to the WT animals (78 ± 6.5 µmol/g) and a consequent increase in feces energy content (WT: 14.442 ± 0.433 kJ/g dry mass compared to dKO: 15.497 ± 0.482 kJ/g dry mass). Thus, lack of TIS7 and SKMc15 reduced efficient free fatty acid uptake in the intestines of mice.

5) It would be important to know if increased MEK/ERK signaling and SOX9 expression are seen in fat pads of mutant mice, not just on the MEF system. Similarly, what are the expression levels of PPARg and C/EBPa in WAT depots of mutant mice?

To address this point, we have now performed the MEK/ERK activity measurement for the revised version of the manuscript in gonadal WAT tissue (GWAT). As noted in samples from several mice, there was an increase in p42 and p44 MAPK phosphorylation in G WAT isolated from dKO mice compared with the G WAT from WT control mice (Figure 4G).

The mRNA expression levels of PPARg and C/EBPa were significantly downregulated in GWAT samples isolated from dKO mice compared with levels from WT control animals (Figure 4H). However, we did not find any significant difference in *SOX9* expression in fat pads. Total amounts of *Sox9* mRNA in terminally differentiated adipocytes were very low and not within the reliable detection range, and the variation between animals within the same group was too great. Therefore, we provide these data only for the reviewer’s information in Author response image 1 and do not present them in the manuscript.

**Author response image 1. sa2fig1:** 

6) Analysis of Wnt signaling in Figure 3c should also include a FOPflash control reporter vector, to demonstrate specificity. Also, data from transfection studies should be shown as mean plus/minus STD and not SEM. This also applies to all other cell-based studies (e.g., Figure 6b,c).

To address the reviewer’s concerns, we performed FOPflash control reporter measurements in MEFs of all four genotypes. As expected, in every tested cell line the luciferase activity of the FOPflash reporter was substantially lower than that of TOPflash, confirming the specificity of this reporter system.

We also thank the reviewer for this important reference to our statistical analyses. We have revised the original data and found that the abbreviation SEM was inadvertently used in the legends instead of STD. STD was always used in the original analyses and therefore we have corrected all legends accordingly in the new version of the manuscript.

7) It is unclear why the authors used the MEF model rather than adipocyte precursors derived from the stromal vascular fraction (SVF) of fat pads from mutant mice. If they did generate data from SVF progenitors, they should include it.

We agree with this comment, although performing the experiments was challenging enough for us. Therefore, we isolated inguinal fat pads and obtained SVF cells from mice of all four genotypes (WT, TIS7, SKMc15 single and double KOs) and have repeated crucial experiments, i.e. adipocyte differentiation, DLK1, PPARg and C/EBPa mRNA and protein analyses in these cells. Novel data gained in this cell system fully confirmed our previous observations in MEFs. Therefore, in the current version of the manuscript we have replaced figures describing the effects of lacking TIS7 and SKMc15 in MEFs by adipose tissues samples (Figures 2D,E, 4G,H,I and 6C) or SVF cells from inguinal WAT (Figures 2A,B,F,G,H, 3C,D,E and F). In addition to the results obtained from SVF cells of inguinal WAT, we also obtained comparable data from SVF cells isolated from fat pads of gonadal WAT. We provide the results from gonadal WAT in Author response image 3 and Author response image 4 for the reviewers' information only.

**Author response image 3. sa2fig3:** 

**Author response image 4. sa2fig4:** 

The only experiments where we have still used data obtained in MEFs are those where the ectopic expression or effects of shRNA were necessary (e.g. Figures 2C, 3B,H,I, 5F,G EV2B,C and EV3 A-F).

8) Given that the authors' proposed mechanism involves both, transcriptional and post-transcriptional regulation of DLK-1 by TIS7 and SKMc15, Figure 4d should be a Western blot capturing both of these events, and not just quantitation of mRNA levels.

As requested by the reviewer, we have added in Figure 3B the Western blot analysis of DLK1 expression. Secondly, this experiment was entirely redone and we now show the effects of ectopic expression of SKMc15, TIS7 alone and their combination side by side with the control GFP. We present here the effects of stable expression of ectopic TIS7 and SKMc15 in dKO MEFs following the viral delivery of expression constructs, antibiotic selection and 8 days of adipocyte differentiation.

9) There is no mention of the impact on brown adipose tissue (BAT) differentiation of KO of TIS7, SKMc15, or the combination. Given the role of BAT in systemic metabolism beyond energy expenditure, the authors need to comment on this issue.

We thank the reviewer for bringing up this important point that prompted us to better describe the phenotype of TIS7, SKMc15 and double knockout mice. We measured DLK1 protein levels in BAT isolated from WT, TIS7, and SKMc15 mice with single and double knockout and detected a significant increase in DLK1 protein levels in all three knockout genotypes. Five mice per genotype were analyzed, and the statistical analysis in Figure 4I represents the mean ± STD. The p-values are based on the results of the Student's t-test and one-way Anova analysis (p-value = 0.0241).

Minor comments:10) The y axis in Figure 2c is labeled as gain of body weight (g). Is it really the case that WT mice gained 30 g of body weight after just 3 weeks of HFD? This rate of increase seems extraordinary, and somewhat unlikely. Please re-check the accuracy of this panel.

We thank the reviewer for drawing our attention to the apparent mislabeling of the y-axis. The correct labeling is: "Increase in body weight in %" and Figure 1F has been corrected accordingly.

11) The Methods indicates all statistical analysis was performed using t tests, but this is at odds with some figure legends that indicate additional tests (e.g., ANCOVA).

This inaccurate information in the manuscript was corrected.

12) Please specify in all cases the WAT depot used for the analysis shown (e.g., Figure 3d is just labeled as WAT, as are Figure 4a,e, etc.).

This information was added at all appropriate places of the manuscript.

13) Figure 5d is missing error bars, giving the impression that this experiment was performed only once (Figure 5c). The legend has no details. Please amend.

We thank the reviewer for this important point regarding the statistical analyses. In the new version of the manuscript, we have included a graph (now Figure 4D) depicting results of three independent experiments including the results of the statistical analysis performed. Statistical analysis was performed using One-Way ANOVA (P=0.0016).

Reviewer #1 (Significance (Required)):The role of TIS7 in adipocyte differentiation is well established. The only truly novel finding in this work is the observation that SKMc15 also plays a role in adipogenesis. The molecular mechanisms proposed (modulation of DLK-1 levels) are not novel, but make sense. However, they need to be bolstered by additional data.Referees cross-commentingI think we are all in agreement that the findings in this work are of interest, but that significant additional work is required to discern the mechanisms involved. In my view, a direct and specific link between SKMc15 and translation of DLK-1 needs to be established and its significance for adipogenesis in cells derived from the SVF of fat pads determined. Reviewer 2 has suggested some concrete ways to provide evidence of a direct link.

We agree with the reviewer's comment and have also noted that this point will be crucial in assessing the novelty value of our manuscript, as was also expressed in the referees cross-commenting. Therefore, we have now additionally performed a polysomal RNA analysis, which has of course been included in the current version of the manuscript.

We analyzed the differences in DLK-1 translation between wild-type control cells and SKMc15 knockout cells in the gradient-purified ribosomal fractions by DLK-1 qPCR. Our analysis identified significantly (p<8,26104E-07) higher DLK-1 mRNA levels in polysomal RNA fraction of SKMc15 KO when compared to wild type MEFs (Figure 3I).

Similarly, as proposed by the reviewer, we have established stromal vascular fraction cell cultures from inguinal fat pads. In SVF cells of TIS7 and SKMc15 single and double knockout mice, we found increased DLK1 mRNA and protein levels (Figures 2F,G and H) as well as decreased PPARg and C/EBPa levels (Figures 3C,D,E and F). Specifically, we found that the ability of knockout SVF cells to differentiate into adipocytes was significantly downregulated (Figures 2A and B), fully confirming our original findings in TIS7 and SKMc15 knockout MEFs.

Reviewer #2 (Evidence, reproducibility and clarity (Required)):Summary:In the current study, Vietor et al. aimed to explore the regulation of Δ-like homolog 1 (DLK-1), an inhibitor of adipogenesis, and demonstrated a role for TIS7 and its orthologue SKMc15 in the regulation of adipogenesis by controlling the level of DLK-1. Using mouse models with whole body deficiency of TIS7 (TIS7 KO) or SKMc15 (SKMc15KO) and double KO (TIS7 and SKMc15 dKO) mice, the authors used a combination of in-vivo experiments and cell culture experiments with mouse embryonic fibroblasts derived from the KO animals, to show that the concurrent depletion of TIS7 and SKMc15 dramatically reduced the amount of adipose tissues and protected against diet-induced obesity in mice, which was associated with defective adipogenesis in vitro.Major Comments:Overall, this study presents convincing evidence that TIOS7 and SKMc15 are necessary for optimal adipogenesis, and proposes a novel mechanism for the control of DLK1 abundance via coordinated regulation of DLK-1 transcription and translation. However, a number of questions remain largely unanswered. In particular, the direct ability of SKMc15 to regulate the translation of DLK-1 is lacking, and this claim remains speculative. SKMc15 being a general inhibitor of translation, SKMc15 may have an effect on adipogenesis independently of its regulation of DLK-1. Thus, addressing the following comments would further improve the quality of the manuscript:

We have been very attentive to these comments to improve the novelty and quality of our manuscript and have tried to address them experimentally. Therefore, this thorough revision of our manuscript took a longer time. First, we identified polysomal enrichment of DLK-1 RNA in SKMc15 KO MEFs, demonstrating that SKMc15 translationally affects DLK-1 levels (Figure 3I). Second, treatment with a recombinant DLK-1 protein as well as its ectopic expression quite clearly blocked adipocyte differentiation of WT MEFs (Figures EV3B,C). In addition, two different shRNA constructs targeting DLK-1 significantly induced adipocyte differentiation of TIS7 SKMc15 dKO MEFs (Figure EV2C, left and right panels). We believe that these results, taken together, sufficiently support our proposed mechanism, namely that TIS7 and SKMc15 control adipocyte differentiation through DLK-1 regulation.

The experimental evidence supporting that SKMc15 controls DLK-1 protein levels comes primarily from the observations that DLK-1 abundance is further increased in SKMc15 KO and dKO WAT than in TIS7KO WAT (Figure 3d), and that translation is generally increased in SKMc15 KO and dKO cells (Figure 5a). However, since the rescue experiment is performed in dKO cells, by restoring both TIS7 and SKMc15 together, it is impossible to disentangle the effects on DLK-1 transcription, DLK-1 translation and on adipogenesis. A more detailed description of the TIS7 and SKM15c single KO cells, with or without re-expression of TIS7 and SKMc15 individually, at the level of DLK-1 mRNA expression and DLK-1 protein abundance would be necessary. In addition, polyribosome fractioning followed by qPCR for DLK-1 in each fraction, and by comparison with DLK-1 global expression in control and SKMc15 KO cells, would reveal the efficiency of translation for DLK-1 specifically, and directly prove a translational control of DLK-1 by SKMc15. Alternatively, showing that DLK-1 is among the proteins newly translated in SKMc15 KO cells (Figure 5a) would be helpful.

As suggested by the reviewer we used single TIS7 and SKMc15 knockout cells and demonstrated that both, TIS7 and SKMc15, affect Dlk-1 mRNA levels. We identified a highly significant effect on total DLK-1 mRNA levels in SKMc15 knockout MEFs as presented in Figure 3H. We also show that DLK-1 mRNA is specifically enriched in polysomal fractions obtained from proliferating SKMc15 knockout MEFs when compared to WT MEFs. However, the strong accumulation of DLK-1 mRNA in polysomes cannot be explained by transcriptional upregulation of DLK-1 alone, suggesting that regulation also occurs at the translational level. We took up this suggestion and ectopically expressed TIS7 and SKMc15 separately or together. For this purpose, we used not only MEF cell lines with double knockout but also with single knockout. Our recent data showed that stable ectopic expression of SKMc15 significantly increased adipocyte differentiation in both, single and double TIS7 and SKMc15 knockout MEF cell lines (Figures EV1C,D and EV2A). Ectopic expression of TIS7 significantly induced the adipocyte differentiation in TIS7 single knockout MEFs (Figure EV1C). In addition, both genes down regulated DLK-1 mRNA expression in dKO MEFs (Figure EV2A, bar chart on the right). We fully agree with the opinion of both reviewers and as already explained above we identified by qPCR in the polysomes that SKMc15 directly regulates DLK-1 translation (Figure 3I).

While the scope of the study focuses on the molecular control of adipogenesis by TIS7 and SKMc15 via the regulation of DLK-1, basic elements of the metabolic characterization of the KO animals providing the basis for this study would be useful. Since the difference in body weight between WT and dKO animals is already apparent 1 week after birth (Figure 1a), it would be interesting to determine whether the fat mass is decreased at an earlier age than 6 months (Figure 1b). The dKO mice are leaner despite identical food intake, activity and RER (Sup Figure 1). It remains unclear whether defective fat mass expansion is a result or consequence of this phenotype. Is the excess energy stored ectopically? The authors mention defective lipid absorption, however, these data are not presented in the manuscript. It would be interesting to investigate the relative contribution of calorie intake and adipose lipid storage capacity in the resistance to diet-induced obesity. In addition, data reported in Figure 1c seem to indicate a preferential defect in visceral fat development, as compared to subcutaneous fat. It would be relevant if the authors could quantify it and comment on it. Are TIS7 and SKMc15 differentially expressed in various adipose depots? The authors used embryonic fibroblasts as a paradigm to study adipogenesis. It would be important to investigate, especially in light of the former comment, whether pre-adipocytes from subcutaneous and visceral stroma-vascular fractions present similar defects in adipogenesis.

We addressed the issue of lipid storage capacity raised by the reviewer using two experimental methods. First, we have analyzed feces of mice fed with high fat diet. The free fatty acids content in dKO mice feces was significantly (P<0.05) higher than in that of their WT siblings (Figure 6A). Second, the energy content of dried feces from dKO mice, as determined by bomb calorimetry, was significantly higher than that from WT mice (P<0.001) (Figure 6B).

Concerning the question of younger animals, we have repeated microCT fat measurements on a group of 1-2 months old WT and dKO male mice (n=4 per group). The total amount of abdominal fat was in WT mice significantly higher than in dKO mice (P=0.019; Student’s T-test). We provide these data only for the reviewer’s information in Author response image 5 and do not present them in the manuscript.

**Author response image 5. sa2fig5:** 

We have also followed the reviewer’s advice and revisited our microCT measurements of abdominal fat and anylyzed the possible differences between subcutaneous and visceral fat. In all three types of abdominal fat mass measurement (total, subcutaneous and visceral) there was always significantly (ANOVA P=0.034 subcutaneous, P=0.002 total and P=0.002 visceral fat) less fat in the dKO group (n=8) of mice when compared to WT (n=12) mice. However, the difference was more prominent in visceral (P=0.001; Student’s T-test) than in subcutaneous fat (P=0.027; Student’s T-test). We provide these data only for the reviewer’s information in Author response image 6 and do not present them in the manuscript.

**Author response image 6. sa2fig6:** 

In addition, we have analyzed the expression of TIS7 and SKMc15 mRNA expression in both, inguinal and gonadal WAT. Our qPCR result showed that both genes are expressed in different types of WAT. The qPCR analysis was performed on RNA isolated from undifferentiated SVF cells isolated from several animals. The expression of TIS7 and SKMc15 was normalized on GAPDH. Data represent mean and standard deviation of technical replicates from several mice as labeled in the graph. We provide these data only for the reviewer’s information in Author response image 7 and do not present them in the manuscript.

**Author response image 7. sa2fig7:** 

Topics of (a) stromal vascular fraction as a source of pre-adipocytes and (b) comparison of TIS7 and SKMc15 roles in gonadal vs. inguinal fat pads we answered in response to the Reviewer #1, point 7. The results are presented in Figures 2, 3 and 4 and in this document.

Both data and methods are explained clearly. The experiments are, for the most part, adequately replicated. However, whenever multiple groups are compared, ANOVA should be employed instead of t-test for statistical analysis.

Thank you for pointing this out. Wherever it was applicable, we used ANOVA for the statistical analysis of data.

Minor comments:Figure 4 d. The appropriate control would be WT with empty vector.

This experiment was entirely replaced by the new Figure 3B where stably transfected MEF cells expressing TIS7 or SKMc15 were used.

Figure 7c/d. The appropriate control would be WT with empty vector.

We have now generated new, confirmatory data in MEF cells stably expressing TIS7 or SKMc15 following lentiviral expression.

Figure 5C. An additional control would be WT with WT medium.

We agree with your suggestion and therefore we have incorporated this control in all experimental repeats presented in the new Figure 4C.

Figure 2: In the legends, the "x" is missing for the dKO regression formula.

Thank you, we have corrected this mistake. In the current version of the manuscript it is Figure 1D.

Since the role of SKMc15 in adipogenesis has never been described, the authors could consider describing the single SKMc15 KO in addition to the dKO, or explain the rationale for focusing the study on dKO.

The original reason for focusing on dKO mice and cells was the obvious and dominant phenotype in this animal model. However, we have sought to address the reviewer's concerns and have now also examined DLK-1 mRNA levels in proliferating SKMc15 knockout MEFs (Figure 3H). In addition to this experiment, we measured DLK-1 mRNA levels also during the process of adipocyte differentiation of single knockout cells. In WT MEFs we observed a transient increase of DLK-1 mRNA only on day 1. In contrast, significantly elevated DLK-1 mRNA levels were found in TIS7 single-knockout MEFs throughout the differentiation process, with the highest level reached at day 8. Interestingly, in SKMc15 single knockout MEFs we found an upregulation of DLK-1 mRNA level in proliferating cells but not a further increase during the differentiation. This supported our idea that SKMc15 acts mainly via translational regulation of DLK-1. We provide these data only for the reviewer’s information in Author response image 8 and do not present them in the manuscript.

**Author response image 8. sa2fig8:** 

To emphasize this point, we revised the entire manuscript accordingly and added data on SKMc15 knockout mice. In particular, experiments presenting data characterizing SKMc15 single knockout mice are presented in: Figures 1C,D,E and F, Figures 2A,B,C and D, Figures 3E,F,H and I, Figures 4A and I and in Figure EV1D.

Reviewer #2 (Significance (Required)):While the effects of DLK-1 on adipogenesis have been widely documented, the factors controlling DLK-1 expression and function remain poorly understood. Here the authors propose a novel mechanism for the regulation of DLK-1, and how it affects adipocyte differentiation. This study should therefore be of interest for researchers interested in the molecular control of adipogenesis and cell differentiation in general. Furthermore, the characterization of the function of SKMc15 in the control of translation may be of interest to a broader readership.Referees cross-commentingI agree with all the comments raised by the other reviewers. Addressing the often overlapping but also complementary questions would help to clarify the molecular mechanisms by which TIS7 and SKMc15 control adipogenesis, and support the conclusions raised by the authors.Reviewer #3 (Evidence, reproducibility and clarity (Required)):In the article, "The negative regulator DLK1 is transcriptionally regulated by TIS7 (IFRD1) and translationally by its orthologue SKMc15 (IFRD2)", the authors performed a double knockout (dKO) of TIS7 and its orthologue SKMc15 in mice and could show that those dKO mice had less adipose tissue compared to wild-type (WT) mice and were resistant to a high fat-diet induced obesity. The study takes advantage of number of different methods and approaches and combines both in vivo and in vitro work. However, some more detailed analysis and clarifications would be needed to fully justify some of the statements. Including the role of TIS7 as a transcriptional regulator of DLK1, SKMc15 as translational regulator of DLK1 and overall contribution of DLK1 in the observed differentiation defects. The observed results could still be explained by many indirect effects caused by the knock-outs and more direct functional connections between the studied molecules would be needed. Moreover, some assays appear to be missing biological replicates and statistical analysis. Please see below for more detailed comments:Major comments:– Are the key conclusions convincing? Yes.– Should the authors qualify some of their claims as preliminary or speculative, or remove them altogether? No.– Would additional experiments be essential to support the claims of the paper? Yes. Please see my comments.– Are the suggested experiments realistic in terms of time and resources? Recombinant DLK1 10 μg – Tetu-bio – 112€ ; 8 days of adipocyte differentiation in 3 biological replicate ~ 1 month.

We followed the advice of the individual reviewers as expressed in “Referees cross-commenting” and tested this idea experimentally. Since the manufacturer couldn’t supply information on biological activities of recombinant DLK-1 proteins, we analyzed in vivo the effects of two different ones, namely RPL437Mu01 and RPL437Mu02. The 8-day adipocyte differentiation protocol showed that the RPL437Mu02 protein was cytotoxic to WT MEF cells and therefore could not be used for analysis. On the other hand, treatment with the Mu01 recombinant DLK-1 protein did not result in a substantial cell death. According to oil red O staining, incubation with 3.3 mg/ml (final concentration) RPL437Mu01 led to 75% inhibition of adipocyte differentiation when compared to not treated WT MEFs (Figure EV3B and C).

– Are the data and the methods presented in such a way that they can be reproduced? Yes.– Are the experiments adequately replicated and statistical analysis adequate? Adequately reproduced yes. Please see my comments concerning the statistical analysis.1) Figure 1A: In the method section it is written that an unpaired 2-tailed Student's t test was used for all statistical comparisons. However, here something like Multivariate analysis of variance (MANOVA) should rather be used to assess statistical significance between the mice. Moreover, the details of this should be clearly stated in the corresponding Figure legend.

Based on this suggestion, we have revised all of our statistical analyses. In several cases, (Figures 1F, 2B and C) we have replaced the statistical analysis using Student’s T test with Anova. However, based on the definition “the difference between ANOVA and MANOVA is merely the number of dependent variables fit. If there is one dependent variable then the procedure ANOVA is used”, in case of Figure 1A we used ANOVA.

2) Figure 2A: please use an appropriate title for Figure 2A instead of "Abdominal fat vs. body mass".

Title of the Figure 1D (formerly Figure 2a) we changed to “Effect of TIS7 and SKMc15 on the abdominal fat mass”.

3) Figure 2C: in the method section it is written that an unpaired 2-tailed Student's t test was used for all statistical comparisons. However, in Figure 2C 4 groups are compared (WT, TIS7 KO, SKMc15 KO and dKO) and thus something like Multivariate analysis of variance (MANOVA) should rather be used to assess statistical significance.

For Figure 1F (formerly Figure 2c), in the revised version of the manuscript, we applied the ordinary one-way ANOVA with Holm-Šidák's multiple comparison test. This analysis gave us statistically even more significant results concerning the difference between WT and dKO mice than previously found by Student's T test. The results in detail were as follows:

**Author response table 1. sa2table1:** Statistical analysis of results presented in Figure 1.

Holm-Šidák's multiple comparisons test	Summary	Adjusted P Value
WT vs. TIS7 KO	**	0,0096
WT vs. SKMc15 KO	*	0,0308
WT vs. dKO	****	<0,0001

4) Figure 2 conclusion: Additive or just showing stronger effect?

We have re-phrased the concluding summary for Figure 1F (formerly Figure 2c). We agree that the precise description of differences found between the weight of single and double knockout animals should be described as “stronger” and not additive effect of knockout of both genes.

5) Figure 3A: the microscope picture for SKMc15 KO shows that cells might have died. Please state the percentage of cell death.

We would like to comment on these concerns of the reviewer as follows: In the image in Figure 3 of the original manuscript, the density of SKMc15 KO MEF cells after the adipocyte differentiation protocol was lower than in the WT control. Regarding the possible cell death, the cells stained with Oil Red O were adherent and alive. The adipocyte differentiation protocol consists of 3 days proliferation and further 5 days of differentiation including three changes of media during which dead cells are washed away and their vitality cannot be checked. However, in the meantime, we have repeated this protocol and the density of SKMc15 knockout MEFs was now not substantially lower than those of controls. Despite the comparable cell density, we have seen a substantial negative effect of the SKMc15 knockout on the adipogenic differentiation ability of these cells. Several examples are shown in Author response image 9.

**Author response image 9. sa2fig9:** 

Importantly, in the current version of our manuscript we replaced MEFs (shown in the former Figure 3a) by SVF cells (Figure 2A in the current manuscript). In these cells we did not see any significant difference in their density after 8 days of the adipocyte differentiation protocol.

6) Figure 3B: It would be informative to additionally observe some of marker genes for adipogenesis and whether all of them are affected.

In our newly established SVF cell lines, derived from inguinal WAT we have confirmed data previously identified in MEFs. As shown in the new Figure 3, PPARg and C/EBPa mRNA levels were downregulated in all knockout SVF cell lines, both undifferentiated (Figures 3C and D) and adipocyte differentiated (Figures 3E and F). On the other hand, DLK-1 mRNA and protein levels, both in undifferentiated (Figures 2F and G) and adipocyte differentiated (Figure 2H) SVF cells were significantly upregulated in dKO cells when compared to WT cells.

7) Figure 3B: instead of using an unpaired 2-tailed Student's t test with proportion, an one-way ANOVA would be more appropriate.

On the recommendation of the reviewer, we applied a simple ANOVA to our new data from SVF cells using the Holm-Šidák test for multiple comparisons. The Anova summary using GraphPad Prism Ver. 9.2 identified statistically highly significant (P value <0,0001) differences between WT and all knockout SVF cells (now Figure 2B).

8) Figure 3C: Same comment as for Figure 3B.

Also, in this experiment (now Figure 2C) we used ordinary one-way ANOVA with Holm-Šidák's multiple comparisons test. The ANOVA summary identified statistically highly significant (P value <0,0001) differences between WT and TIS7 single and dKO MEF cells. On the other hand, there was no statistically significant difference between WT and SKMc15 knockout MEFs.

9) Fig3d: A representative Western blot for 3 independent experiments is shown. Please add the other two as supplementary materials.

In Author response image 10 we provide examples of the requested two additional, independent experiments. These refer now to the Figure 2D in the revised version of the manuscript:

**Author response image 10. sa2fig10:** 

10) Fig3d: Is this distinguishing between the active and inactive catenin?

No, the b-catenin antibody, that we used is not discriminating between active and inactive b-catenin forms.

11) Figure 4A: Please perform qPCR for measuring DLK-1 mRNA levels in TIS7 KO and SKMc15 KO samples to check whether there is a correlation between mRNA and protein level as the statement of the authors is that "DLK1 is transcriptionally regulated by TIS7 (IFRD1) and translationally by its orthologue SKMc15".

Similar questions were raised by Reviewer 2 on p. 11 “*Since the role of SKMc15 in adipogenesis has never been described, the authors could consider describing the single SKMc15 KO in addition to the dKO, or explain the rationale for focusing the study on dKO.*” Please see our reply to his comment.

**Author response image 11. sa2fig11:** 

12) Figure 4C: please add the other two western blots as supplementary materials.

In Author response image 12 we provide data from two additional, independent experiments.

**Author response image 12. sa2fig12:** 

13) Fig4d: The effects in MEFs appear quite modest. What about a rescue with TIS7 or SKMc15 alone?

As mentioned already in response to the question 2 of Reviewer #1, in our newly performed experiments we found significant inhibitory effects of ectopic TIS7 and SKMc15 expression on DLK1 levels, identified both by qPCR and WB analyses (Figure 3B).

14) Page 12, row 207: I would not call histones transcription factors.

We re-phrased this sentence accordingly.

15) Fig4e: Would be good to see a schematic overview of the locations of the ChIP primers in relation to the known binding sites and the gene (TSS, gene body). Moreover, the results include an enrichment for only one region while in the text two different regions are discussed. Importantly, to confirm the specificity of the observed enrichment, a primer pair targeting an unspecific control region not bound by the proteins should be included.

The selection of oligonucleotide sequences used for ChIP analyses of the binding of b-catenin, TIS7 and SKMc15 to the Dlk-1 promoter was, based on the following reference, as mentioned in Methods section of our original manuscript on p.21, line 494: Paul C, Sardet C, Fabbrizio E. “The Wnt-target gene Dlk-1 is regulated by the Prmt5-associated factor Copr5 during adipogenic conversion”. Biol Open. 2015 Feb 13;4(3):312-6. doi: 10.1242/bio.201411247.

We used two regions of the Dlk-1 promoter: a proximal one, encompassing the TCF binding site 2 (TCFbs2) and a more distal one, annotated as “A”:

Oligonucleotide sequences used for ChIP PCR:Dlk-1 TCFbs2 5'f CATTTGACGGTGAACATATTGG

5'r GCCCAGACCCCAAATCTGTC

Dlk-1 region A (-2263/-2143) 5'f TTGTCTAACCACCCTACCTCAAA

5’r CTCTGAGAAAAGATGTTGGGATTT

We observed specific binding at the proximal site.

16) Figure 5A: Has this experiment been replicated? That is no mention about the reproducibility or quantification of this result. This is the main experiment regarding the role of SKMc15 as a translational regulator of DLK1, also mentioned in the title of the manuscript.

This relates to the Figure 4A in the revised manuscript. Yes, we repeated this experiment several times. Here we provide images and quantifications of three independent experiments.

17) Figure 5B: Showing another unaffected secreted protein would be an appropriate control here.

As recommended by the reviewer, we have performed an additional WB with a recombinant anti-Collagen I antibody [Abcam, [EPR22209-75] ab255809]. Medium from 8 days adipocyte differentiated WT and dKO MEFs was concentrated using Centriprep 30K and resolved on 10% SDS-PAGE gel. Western blot presented in the new Figure 4 B shows even slightly higher amounts of Collagen-1 protein in medium from WT than in dKO MEFs. Mw of the detected band was approximately 35 kDa, which corresponded to the manufacturer’s information.

18) Figure 5C: I would recommend to perform additional experiments to prove that DLK-1 secreted in the medium can contribute and is responsible for the inhibition of the differentiation. Indeed, a time course of adipocyte differentiation followed by the addition of soluble DLK-1 would confirm that DLK-1 can inhibit adipocyte differentiation in this experimental setup. Moreover, silencing (for example RNAi) of DLK1 in the dKO cells before harvesting the conditioned media would allow to estimate the contribution of DLK1 to the observed inhibition of differentiation by the media. This is important because many other molecules could also be mediating this inhibition.

We agree with this reviewer’s concern, which are shared by other reviewers. Similarly, as in response to Reviewer #2 and as already mentioned above, in response to “major comments” of Reviewer #3, in our novel experiments we found that treatment with recombinant DLK-1 protein as well as ectopic expression of DLK-1 blocked adipocyte differentiation of WT MEFs (Figures EV3B,C,D and E) as well as medium from dKO shDLK-1 391 cells (Figure EV3F).

19) Figure 5C: The details and the timeline of the experiment with conditioned media are not provided in the figure or in the methods. At what time point was conditioned media changed? How long were the cells kept in conditioned media? How does this compare to the regular media change intervals? Could the lower differentiation capacity relate to turnover of the differentiation inducing compounds in the media due to longer period between media change? Moreover, is the result statistically significant after replication?

Based on the reviewer`s comment we have added technical information concerning the experimental protocol of the treatment with conditioned media. In general, the treatment for adipocyte differentiation was identical with the previous experiments. The only difference was that after three days in proliferation medium, we used either fresh differentiation medium or 2-day-old differentiation medium from dKO control or dKO-shDLK-1 391 cell cultures then for wild-type cells, as shown in the figure (Figure EV3F). Cells were incubated additional five days with the differentiation medium with two changes of media, every second day. The adipocyte differentiation of medium “donor” cells and the DLK-1 protein levels in these cells were monitored by oil red O staining and Western blot analysis, respectively.

Additionally, we show now in Figure 4C representative images from three independent biological repeats and in Figure 4D the statistical analysis confirming a significant decrease in adipocyte differentiation ability of WT MEFs following their incubation with a conditioned differentiation medium from dKO MEFs.

20) Fig5d: please add a statistical analysis of the oil-red-o quantification.

As requested, we included statistical analysis of at least three independent experiments. In Figure 4D we present the statistical analysis confirming a significant decrease in adipocyte differentiation of WT MEFs following their incubation with the differentiation medium from dKO cells. Additionally, Figure 4C shows representative images of oil red O staining from several independent experiments.

21) Fig7c-d: Does overexpression also rescue the PPARg and CEBPa induction during differentiation. The importance of their induction in undifferentiated MEFs is a little difficult to judge.

We have focused our attention primarily on the ability of TIS7 and SKMc15 to “rescue” the adipocyte differentiation phenotype of dKO MEFs. dKO MEFs stably expressing SKMc15, TIS7 or both genes were differentiated into adipocytes for 8 days and afterwards stained with oil red O. There was a statistically significant increase in oil red O staining following the individual ectopic expression of SKMc15 (p=5.7E-03), a negative effect of TIS7 ectopic expression and a significant (p=9.3E-03), positive effect of co-expression of both genes (Figure EV2A). We found a significant decrease in Dlk-1 mRNA expression following the ectopic expression of TIS7 and/or SKMc15 (Figure EV2A, very right panel). However, C/EBPa mRNA levels were only partially rescued in 8 days differentiated MEFs by TIS7 and/or SKMc15 ectopic expression, and PPARg mRNA levels were not significantly altered.

22) Fig8: it is not surprising that PPARg targets are not induced in the absence of PPARg. What is the upstream event explaining this defect? Is DLK1 alone enough to explain the results? Could there be additional mediators of the differences? How big are transcriptome-wide differences between WT MEFs and dKO MEFs?

We agree with the reviewer that the lean phenotype of dKO mice most likely cannot be explained by simple transcriptional regulation of PPARg. Although we showed that in undifferentiated MEFs, the levels of PPARg and C/EBPa are controlled (or upregulated) by both TIS7 and SKMc15, we also expected differences in the expression of genes regulating fat uptake. To determine changes in expression of lipid processing and transporting molecules, we performed transcriptome analyses of total RNA samples isolated from the small intestines of HFD-fed WT type and dKO animals. Cluster analyses of lipid transport-related gene transcripts revealed differences between WT type and dKO animals in the expression of adipogenesis regulators. Those included among other genes the following, mentioned as examples:

peroxisome proliferator-activated receptors γ (PPARγ) and d [2], fatty acid binding proteins 1 and 2 (FABP1, 2) [3],cytoplasmic fatty acid chaperones expressed in adipocytes,acyl-coenzyme A synthetases 1 and 4 (ACSL1,4) found to be associated with histone acetylation in adipocytes, lipid loading and insulin sensitivity [4],SLC27a1, a2 fatty acid transport proteins, critical mediators of fatty acid metabolism [5],angiotensin-converting enzyme (ACE) playing a regulatory role in adipogenesis and insulin resistance [6],CROT, a carnitine acyltransferase important for the oxidation of fatty acids, a critical step in their metabolism [7],phospholipase PLA2G5 robustly induced in adipocytes of obese mice [8]; [9].

Parts of the following text are embedded in the manuscript.

We decided to study in more detail the regulation of CD36 that encodes a very long chain fatty acids (VLCFA) transporter because CD36 is an important fatty acid transporter that facilitates fatty acids (FA) uptake by heart, skeletal muscle, and also adipose tissues [10]. PPARγ induces CD36 expression in adipose tissue, where it functions as a fatty acid transporter, and therefore, its regulation by PPARγ contributes to the control of blood lipids. Diacylglycerol acyltransferase 1 (DGAT1), a protein associated with the enterocytic triglyceride absorption and intracellular lipid processing is besides CD36 another target gene of adipogenesis master regulator PPARγ [11]. DGAT1 mRNA levels are strongly up regulated during adipocyte differentiation [12], its promoter region contains a PPARγ binding site and DGAT1 is also negatively regulated by the MEK/ERK pathway. DGAT1 expression was shown to be increased in TIS7 transgenic mice [13] and its expression was decreased in the gut of high fat diet-fed TIS7 KO mice [14]. Importantly, DGAT1 expression in adipocytes and WAT is up regulated by PPARγ activation [11].

**Author response image 13. sa2fig13:** Heatmap of hierarchical cluster analysis of intestinal gene expression involved in lipid transport altered in TIS7 SKMc15 dKO mice fed a high-fat diet for 3 weeks.

What is the upstream event explaining this defect?

Wnt pathway causes epigenetic repression of the master adipogenic gene PPARγ. There are three epigenetic signatures implicated in repression of PPARγ: increased recruitment of MeCP2 (methyl CpG binding protein 2) and HP-1α co-repressor to PPARγ promoter and enhanced H3K27 dimethylation at the exon 5 locus in a manner dependent on suppressed canonical Wnt. These epigenetic effects are reproduced by antagonism of canonical Wnt signaling with Dikkopf-1.

Zhu et al. showed that Dlk1 knockdown causes suppression of Wnt and thereby epigenetic de-repression of PPARγ [15]. Dlk1 levels positively correlate with Wnt signaling activity and negatively with epigenetic repression of PPARγ [16]. Activation of the Wnt pathway caused by DLK1 reprograms lipid metabolism via MeCP2-mediated epigenetic repression of PPARγ [17]. Blocking the Wnt signaling pathway abrogates epigenetic repressions and restores the PPARγ gene expression and differentiation [18].

Minor comments:1) Please use the same font in the main text for the references.

We thank the reviewer for the remark. This issue was corrected.

Reviewer #3 (Significance (Required)):The study provides interesting insights into the role of these factors in adipocyte differentiation that would be relevant especially to researchers working on adipogenesis and cellular differentiation in general. The authors find the studied factors to have additive contribution to the differentiation efficiency. However, the exact nature of the roles and whether they are strictly speaking additive or synergistic is not clear. More detailed analysis of their contribution and molecular interplay would add to the broader interest of the study on molecular networks controlling cellular differentiation.Referees cross-commentingI very much agree on the different points raised by the other reviewers, some of which are also matching my own already raised concerns. And therefore it makes sense to request these modifications from the authors.

References

1. Rozman, J., M. Klingenspor, and M. Hrabe de Angelis, *A review of standardized metabolic phenotyping of animal models.* Mamm Genome, 2014. 25(9-10): p. 497-507.

2. Lefterova, M.I., et al., *PPARgamma and the global map of adipogenesis and beyond.* Trends Endocrinol Metab, 2014. 25(6): p. 293-302.

3. Garin-Shkolnik, T., et al., *FABP4 attenuates PPARgamma and adipogenesis and is inversely correlated with PPARgamma in adipose tissues.* Diabetes, 2014. 63(3): p. 900-11.

4. Joseph, R., et al., *ACSL1 Is Associated With Fetal Programming of Insulin Sensitivity and Cellular Lipid Content.* Mol Endocrinol, 2015. 29(6): p. 909-20.

5. Anderson, C.M. and A. Stahl, *SLC27 fatty acid transport proteins.* Mol Aspects Med, 2013. 34(2-3): p. 516-28.

6. Riedel, J., et al., *Characterization of key genes of the renin-angiotensin system in mature feline adipocytes and during* in vitro *adipogenesis.* J Anim Physiol Anim Nutr (Berl), 2016. 100(6): p. 1139-1148.

7. Zhou, S., et al., *Increased missense mutation burden of Fatty Acid metabolism related genes in nunavik inuit population.* PLoS One, 2015. 10(5): p. e0128255.

8. Wootton, P.T., et al., *Tagging SNP haplotype analysis of the secretory PLA2-V gene, PLA2G5, shows strong association with LDL and oxLDL levels, suggesting functional distinction from sPLA2-IIA: results from the UDACS study.* Hum Mol Genet, 2007. 16(12): p. 1437-44.

9. Sergouniotis, P.I., et al., *Biallelic mutations in PLA2G5, encoding group V phospholipase A2, cause benign fleck retina.* Am J Hum Genet, 2011. 89(6): p. 782-91.

10. Coburn, C.T., et al., *Defective uptake and utilization of long chain fatty acids in muscle and adipose tissues of CD36 knockout mice.* J Biol Chem, 2000. 275(42): p. 32523-9.

11. Koliwad, S.K., et al., *DGAT1-dependent triacylglycerol storage by macrophages protects mice from diet-induced insulin resistance and inflammation.* J Clin Invest, 2010. 120(3): p. 756-67.

12. Cases, S., et al., *Identification of a gene encoding an acyl CoA:diacylglycerol acyltransferase, a key enzyme in triacylglycerol synthesis.* Proc Natl Acad Sci U S A, 1998. 95(22): p. 13018-23.

13. Wang, Y., et al., *Targeted intestinal overexpression of the immediate early gene tis7 in transgenic mice increases triglyceride absorption and adiposity.* J Biol Chem, 2005. 280(41): p. 34764-75.

14. Yu, C., et al., *Deletion of Tis7 protects mice from high-fat diet-induced weight gain and blunts the intestinal adaptive response postresection.* J Nutr, 2010. 140(11): p. 1907-14.

15. Zhu, N.L., et al., *Hepatic stellate cell-derived δ-like homolog 1 (DLK1) protein in liver regeneration.* J Biol Chem, 2012. 287(13): p. 10355-10367.

16. Zhu, N.L., J. Wang, and H. Tsukamoto, *The Necdin-Wnt pathway causes epigenetic peroxisome proliferator-activated receptor γ repression in hepatic stellate cells.* J Biol Chem, 2010. 285(40): p. 30463-71.

17. Tsukamoto, H., *Metabolic reprogramming and cell fate regulation in alcoholic liver disease.* Pancreatology, 2015. 15(4 Suppl): p. S61-5.

18. Miao, C.G., et al., *Wnt signaling in liver fibrosis: progress, challenges and potential directions.* Biochimie, 2013. 95(12): p. 2326-35.

[Editors' note: further revisions were suggested prior to acceptance, as described below.]

Based on the previous reviews and the revisions, the manuscript has been improved but there are some remaining issues that need to be addressed, as outlined below:This study provides new insights into the molecular regulation of adipocyte differentiation. Two molecules, TIS7 and SKMc15, are shown to regulate the activity of the key transcriptional regulator DLK-1 via discrete mechanisms – one involving transcription and the other translation. These findings add additional information to the well known roles of Wnt/catenin and PPARg on adipocyte differentiation.The authors have extensively addressed the comments of the referees and all referees are convinced that the manuscript is now improved and the mechanistic details of the link between SKMc15 and regulation of adipogenesis is now strengthened. However, as described below there are still some important issues that need to be addressed by the authors prior to publication. Many of these require editorial changes but there are some minor experimental details that need to be addressed. Also essential is that you discuss more thoroughly the lipid absorption issue as a contributor to the dKO mouse phenotype.Specific Issues1) All energy balance measurements need to be included in the manuscript, not just shown to reviewers. Readers will want to see them.

We added results of the indirect calorimetry trial to the manuscript as Table 1. Text changes are in lines 179-181 and the new legend is 973-978.

2) It is not quite clear how food intake is expressed. Since the mice are significantly smaller, it might be more appropriate to express the data as g of food/g of mouse, as a smaller mouse is likely to eat less.

This issue is addressed in Table 1. Four parameters, among them also food intake, were first statistically analyzed using one-way ANOVA and secondly using a linear model including body mass as a co-variate, meaning normalized on the body mass. This parameter was graphically presented in the Figure 1 —figure supplement 2A. In the legend to this figure (lanes 766-768) is explained that food intake, when adjusted to the body mass, was non-significantly decreased in dKO mice (4.0±0.4 g in WT vs. 3.4±0.5 g in dKO mice).

3) In complementation experiments, it would be useful to know the levels of overexpression.

Western blots of cell lysates from both wild type and SKMc15 or TIS7 over-expressing dKO MEFs were quantified using Vilber spectral imager hardware and software. Supplementary figure I depicts samples from dKO MEFs over-expressing TIS7 and figure J dKO MEFs over-expressing SKMc15. The over-expression levels were 14.7-fold for TIS7 and 10.91-fold for SKMc15, respectively in comparison to the endogenous levels of these proteins in wild type MEF cells. Images shown in Figure 1 —figure supplement 2I and J originate from the same membrane, just lanes with unrelated samples are here not shown.

4) What is the levels of blood lipids in the dKO mice, since this is the first report of their existence, and presumably the defect in intestinal lipid absorption may affect these?

In order to answer this question we have now measured cholesterol, triglycerides levels, and the lipoprotein profile in four wild type and four TIS7 SKMc15 dKO, 7 months old, male mice. None of these three analyses identified any statistically significant difference between wild type and dKO mice. Results of these measurements are presented in the Figure 1 —figure supplement 2K, L, and M and explained in the legend, lanes 798-806.

5) The methods need to be carefully edited. For instance, lines 564-65 state "Small intestines were harvested for oil red O staining to detect lipid accumulation"; these data are not shown anywhere in the paper. Similarly, the section on fecal fat determination describes a protocol to analyze neutral sterols and bile acids (nowhere in the paper), but it does not describe how the free fatty acid levels were determined. One more example, line 651 alludes to "pulse labeling of mitochondrial proteins" which is not what is presented in the paper.

We thank the editor for the careful reading of our manuscript. We corrected all mentioned inaccuracies in the Methods section and completely replaced a paragraph describing the analysis of fatty acids. The new version is now explained in lines 554-558. Plasma cholesterol and triglyceride analyses are described in lines 562-565.

6) The main text should indicate at the outset that only male mice were analyzed.

In the Results section, line 154-155 and in the Material and methods section, lines 496-497 we explicitly note that: “In order to achieve maximal homogeneity of experimental groups, in all experiments presented here we used only male mice”.

7) It is incorrect to state that the dKO mice are not smaller when the graph showing that data (panel A in Figure EV1) shows a significant difference.

This information was corrected, see lines 162-164.